# Tissue-intrinsic beta-catenin signals antagonize Nodal-driven anterior visceral endoderm differentiation

Sina Schumacher[1], Max Fernkorn[1], Michelle Marten[1], Rui Chen[2], Yung Su Kim[2,3], Ivan Bedzhov [2] & Christian Schröter [1] ✉

The anterior-posterior axis of the mammalian embryo is laid down by the anterior visceral endoderm (AVE), an extraembryonic signaling center that is specified within the visceral endoderm. Current models posit that AVE differentiation is promoted globally by epiblast-derived Nodal signals, and spatially restricted by a BMP gradient established by the extraembryonic ectoderm. Here, we report spatially restricted AVE differentiation in bilayered embryo-like aggregates made from mouse embryonic stem cells that lack an extraembryonic ectoderm. Notably, clusters of AVE cells also form in pure visceral endoderm cultures upon activation of Nodal signaling, indicating that tissue-intrinsic factors can restrict AVE differentiation. We identify β-catenin activity as a tissue-intrinsic factor that antagonizes AVE-inducing Nodal signals. Together, our results show how an AVE-like population can arise through interactions between epiblast and visceral endoderm alone. This mechanism may be a flexible solution for axis patterning in a wide range of embryo geometries, and provide robustness to axis patterning when coupled with signal gradients.

Identifying cell-cell communication mechanisms that orchestrate the self-organized development of the mammalian embryo is a major goal in developmental biology. The modularity of stem cell-based embryo-like models offers the possibility to investigate cell differentiation in subsystems, and thereby to identify signaling mechanisms that may have remained hidden in the embryo.

One of the most fundamental processes in embryonic development is the establishment of an anterior-posterior axis. In mammals, this axis is laid down by the anterior visceral endoderm (AVE), a specialized extraembryonic cell population within the visceral endoderm (VE) that overlies the embryonic epiblast at the time of implantation. The AVE expresses transcription factors such as Otx2, Eomes, Gsc and Lhx1, and Wnt, BMP and Nodal antagonists such as Dkk1, Cer1, and Lefty1[1–3]. These secreted signaling antagonists pattern the epiblast by restricting Wnt, BMP and Nodal signaling to its posterior end, thereby establishing the anterior-posterior axis of the embryo.

In rodents, the VE and the epiblast form a cup-shaped egg cylinder (Fig. 1a). The precursor cells of the AVE initially differentiate from the VE at the distal tip of the egg cylinder, before migrating towards the future anterior side. AVE differentiation is promoted by Nodal signals from the epiblast[4], and thought to be locally restricted by graded inhibitory BMP signals from the extraembryonic ectoderm (ExE), an extraembryonic tissue at the proximal end of the egg cylinder[3,5,6]. Cell populations with a similarity to the mouse AVE have been described in non-rodent mammals, including humans[7,8]. Embryos from these species are disc- rather than cup-shaped, and may therefore lack the BMP gradient present in rodent embryos. This raises the possibility that alternative mechanisms for AVE differentiation exist that may be obscured by the activity of graded BMP signals in rodent embryos.

[1]Department of Systemic Cell Biology, Max Planck Institute of Molecular Physiology, Dortmund, Germany. [2]Embryonic Self-Organization research group, Max Planck Institute for Molecular Biomedicine, Münster, Germany. [3]Present address: Integrated Biosystems and Biomechanics Laboratory, Department of Mechanical Engineering, University of Michigan, Ann Arbor, MI, USA. ✉e-mail: christian.schroeter@mpi-dortmund.mpg.de

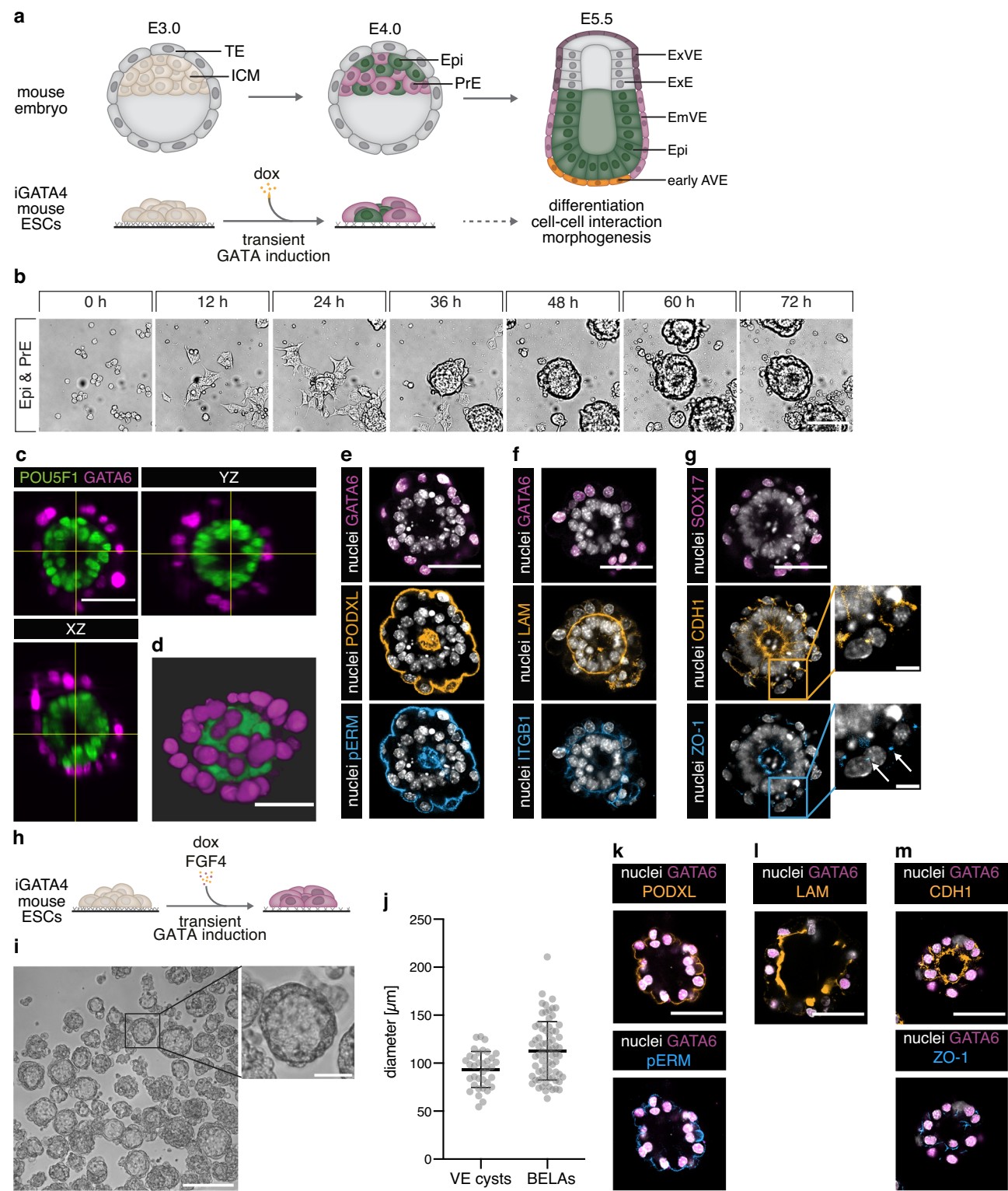

Such alternative mechanisms can be identified with stem cell-based embryo models composed of embryonic and specific extra-embryonic lineages. Here, we use an embryo-like model system consisting of the epiblast and the VE compartment to study how cell-cell communication controls AVE differentiation in the absence of an ExE. We first characterize bilayered aggregates made from mouse embryonic stem cells (ESCs) that recapitulate the interaction of the epiblast and the VE lineage as seen in the embryo, and contrast them with 3D structures that consist of either of the two cell types alone. Using single-cell RNA-sequencing, we show that the presence of the epiblast compartment suffices to trigger differentiation of a subset of VE cells towards an AVE identity. We apply cell-cell communication analysis to identify the associated signaling pathways, and use this knowledge to develop protocols for AVE differentiation in the absence of an epiblast compartment. Stimulation of Activin/Nodal signaling, coupled to inhibition of β-catenin transcriptional activity allows us to differentiate almost pure populations of AVE cells in vitro, suggesting that tissue-intrinsic signals regulating β-catenin activity restrict AVE differentiation to local cell clusters. The combination of signaling mechanisms for AVE differentiation described

**Fig. 1 | Formation and characterization of BELAs and VE cysts. a** Schematic of mouse embryonic development from E3.0 to E5.5 (top) and GATA4-inducible embryonic stem cell (ESC) system to model interactions between epiblast (Epi) and extraembryonic endoderm (bottom). **b** Stills from a movie of ESC-derived Epi and primitive endoderm (PrE) cells seeded on a low adhesion substrate in N2B27 medium. See also Supplementary Movie 1. One representative out of $n = 3$ independent experiments shown. **c, d** Orthogonal views (**c**) and 3D volume rendering (**d**) of a bilayered aggregate imaged with light sheet microscopy. POU5F1 (green) marks Epi identity and GATA6 (magenta) marks PrE/VE identity. See also Supplementary Movie 2. One representative out of $n = 5$ structures shown. **e–g** Immunostainings of bilayered aggregates for the PrE/VE markers GATA6 (**e**), (**f**) or SOX17 (magenta, **g**), the apical markers PODXL (orange) and pERM (blue) (**e**), the

basement membrane and adhesion markers LAM (orange) and ITGB1 (blue) (**f**), and the epithelial markers CDH1 (orange) and ZO-1 (blue) (**g**). One representative out of at least $n = 7$ structures shown. Arrows in (**g**, inset) mark punctate ZO-1 staining characteristic for tight junctions. **h** Schematic of experimental protocol to differentiate pure populations of PrE cells. **i** VE cysts formed in N2B27 supplemented with FGF4 on a low adhesive substrate. One out of $n = 3$ independent experiments shown. **j** Diameters of detached BELAs and VE cysts grown for 3 days on a low adhesive substrate. $n = 72$ (BELAs) and $n = 36$ (VE cysts); bars indicate mean ± SD. **k–m** Immunostainings of VE cysts for the same markers as in **e–g**. One representative out of at least $n = 7$ structures shown. Scale bars: 50 μm in (**b–g**, **i** (inset) and **k–m**), 10 μm in **g** (inset), 200 μm in **i**. Source data for **j** are provided in the Source Data file.

here may help explain axis patterning in embryos that do not have a BMP gradient.

## Results

### Generation of simplified 3D models of the Epi- and VE-compartments

To generate a 3D model of the peri-implantation embryo that contains its epiblast and VE-compartment we started from GATA4-inducible mouse ESCs. We have previously shown that following transient GATA4 expression, these cells differentiate into robust proportions of epiblast (Epi) and primitive endoderm (PrE) cells, the precursors of the VE (Fig. 1a)[9,10]. To promote cell-cell interactions, we re-seeded these cell type mixtures after 16 h and lowered the adhesiveness of the substrate. Under these conditions, cells quickly aggregated, formed round structures consisting of two layers of cells that surrounded a central lumen, and eventually detached from the culture surface (Fig. 1b; Supplementary Movie 1). The outer layer of these spherical structures consisted of GATA6-positive VE cells, while the inner layer expressed the Epi marker POU5F1 (Fig. 1c, d and Supplementary Movie 2). Staining with the apical markers PODXL and pERM showed that both compartments were polarized, with the apical domain of the VE pointing towards the outside of the aggregates, and the apical domain of the Epi layer pointing to the inside (Fig. 1e). At their basal sides, we detected expression of β1-integrin (ITGB1) as well as a laminin-rich basal membrane (Fig. 1f and Supplementary Movie 3). Both layers stained positive for the epithelial markers E-Cadherin (CDH1) and ZO-1 (Fig. 1g). This architecture of two apposed epithelial layers resembles the arrangement of cells in the distal part of the egg cylinder. We therefore term these structures "bilayered embryo-like aggregates" (BELAs).

We compared BELAs to previously described embryoids made from ESCs and extraembryonic endoderm (XEN) cells (EXE embryoids)[11]. EXE embryoids made by mixing ESCs with established XEN cell lines under the same conditions that were used to make BELAs had an overall similar size and architecture as BELAs, but did not detach from the surface of the culture vessel (Supplementary Fig. 1a, b). Furthermore, the outer XEN cell-derived layer in EXE embryoids lacked detectable CDH1 expression as well as the continuous apical domain marked by pERM and PODXL that was commonly seen in BELAs (Supplementary Fig. 1b–d), suggesting that the ESC-derived VE cells in BELAs have a stronger propensity to epithelialize than XEN cells.

We next sought to generate 3D structures that consist of Epi and VE cells in isolation. To obtain only Epi cells, we cultured ESCs under the same conditions as used for BELA formation, but omitted the doxycycline pulse. Under these conditions we observed extensive cell death from 48 h after re-seeding onwards (Supplementary Fig. 2a–c). As previously described, culture of cells in matrigel rescued their survival and induced cyst formation (Supplementary Fig. 2d)[12], suggesting that a major function of the VE layer in BELAs is to provide survival and patterning signals via the extracellular matrix.

To generate pure cultures of PrE cells, we extended the expression of the inducible transgene for 16 h after the switch to N2B27 medium, and supplemented the medium with exogenous FGF4 (Fig. 1h, Supplementary Fig. 2a)[9]. When these cultures were re-seeded on low-adhesion substrates in N2B27 only, we observed rapid cell death (Supplementary Fig. 2b, c), but survival could be fully rescued by continued addition of FGF4 (Supplementary Fig. 2b). Surprisingly, in the presence of FGF4, these cells aggregated and formed non-adherent 3D structures (Fig. 1i and Supplementary Fig. 2b). These aggregates varied in size and shape, but a large number of them formed round cysts with a big lumen (Fig. 1i). The diameter of these cysts was 93.4 ± 18.8 μm (mean ± SD), similar to that of BELAs (112.8 ± 30.4 μm, Fig. 1j). The apical markers PODXL and pERM localized to the outside of the cysts, laminin was secreted to their inside, and the localized expression of CDH1 and ZO-1 further indicated an epithelial organization (Fig. 1k–m). Thus, these structures resemble the outer layer of BELAs, and we hence refer to them as VE cysts. Cyst formation was specific to ESC-derived VE cells, since XEN cells cultured under the same conditions continued growing as a monolayer (Supplementary Fig. 2e).

Taken together, the exchange of mutual survival signals between Epi and PrE cells underlies the spontaneous formation of BELAs. Replacing these signals with purified factors allows us to generate Epi- and VE cysts that consist of only one of the cell types found in BELAs, but that capture the 3D organization of the single compartments.

### Interactions between Epi and VE cells in BELAs shape cell differentiation trajectories

In the post-implantation embryo, differentiation of both the VE as well as the Epi lineage is strongly influenced by signals from the ExE. We reasoned that the three simplified 3D models could reveal mechanisms for cell differentiation that are independent of the ExE. We, therefore, performed single-cell RNA-sequencing (scRNAseq) on the three types of aggregates (Fig. 2a). Representation of the single-cell transcriptomes in a UMAP plot showed two major groups, one containing mainly cells from Epi cysts and a subgroup of the BELA cells. The other group contained the remaining BELA cells as well as most cells from VE cysts (Fig. 2b). Expression of the VE marker genes *Sox17, Cubn, Dab2*, and *Gata6*[12] and the Epi marker genes *Fgf4, Nanog, Pou5f1*, and *Sox2*[13] identified these two broad groups as VE and Epi, respectively (Fig. 2c). To determine which cell types and developmental stages were captured in the in vitro samples, we integrated our scRNAseq data with single-cell transcriptomes from the embryo. We chose a reference dataset that covered several embryonic stages between E3.5 and E8.75, and that focused in particular on the emergence of the endoderm lineage[14]. UMAP representations after integration indicated that cells from BELAs and cysts corresponded to a range of embryo cell types (Fig. 2d, e).

We transferred cell type and stage labels from the reference dataset and plotted their frequency in each sample (Fig. 2f). While cells from BELAs mapped to both E6.5 and E7.5 reference cells, the majority

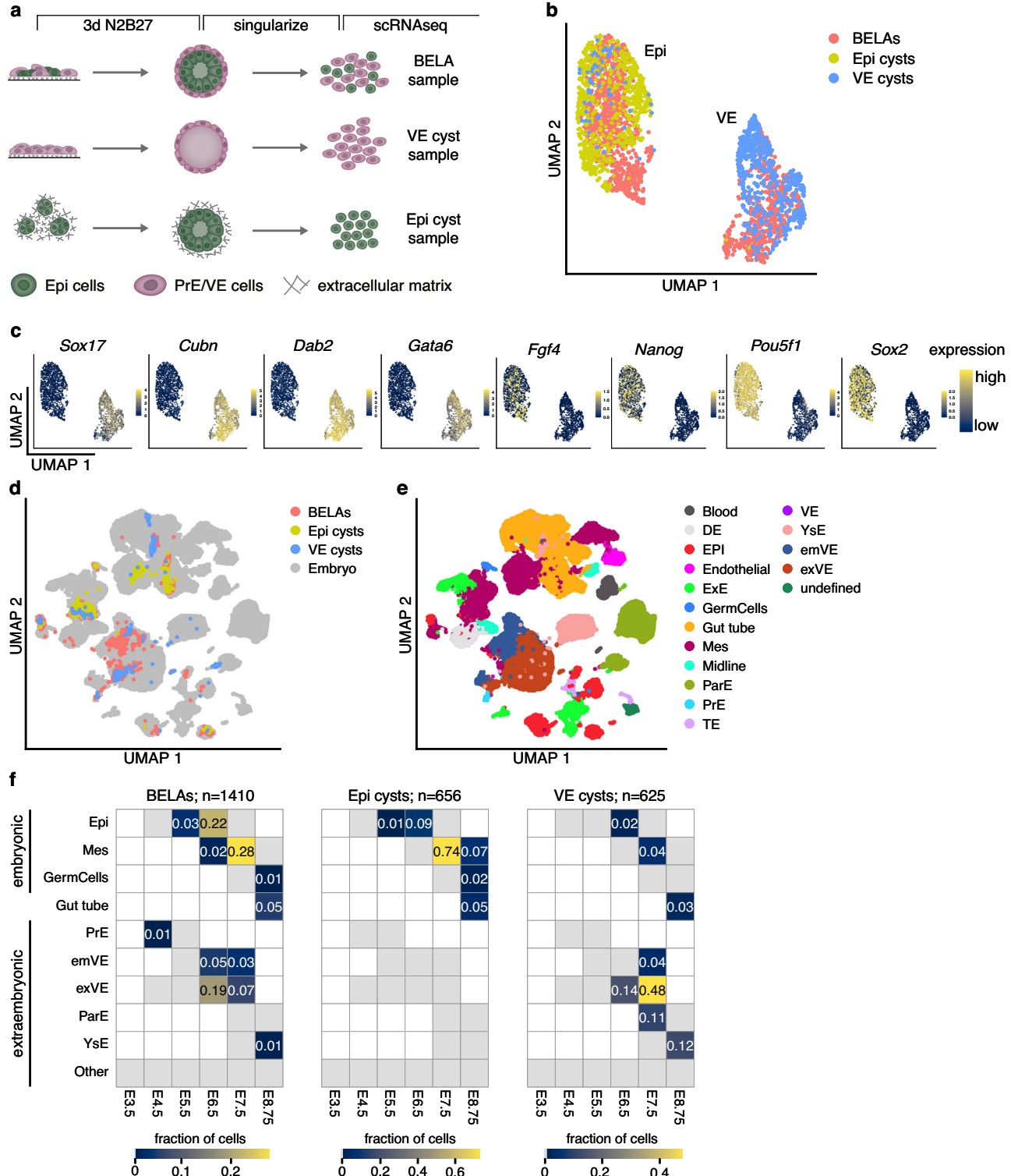

**Fig. 2 | Single-cell RNA-sequencing and data integration to determine cell type identities in BELAs and cysts. a** Experimental approach to prepare samples for scRNAseq. **b** UMAP of batch corrected single-cell transcriptomes from cells prepared as in **a**. Colors indicate sample of origin. **c** Expression levels of VE markers *Gata6, Sox17, Dab2,* and *Cubn*, and Epi markers *Pou5f1, Sox2, Nanog,* and *Fgf4*. To better visualize the cell type-specific expression of *Fgf4, Nanog* and *Sox2*, expression levels above ln ≥1.5 (*Fgf4*) or ln ≥2 (*Nanog* and *Sox2*) are shown in yellow. **d** UMAP of single-cell transcriptomes from BELAs, Epi cysts and VE cysts, integrated with scRNAseq data from mouse embryos covering stages E4.5 to E8.75[14]. **e** Same UMAP as in **d**, colored according to cell type annotation from[14] after integration and label transfer. **f** Heatmaps showing the fraction of cells in BELAs (left), Epi cysts (middle) and VE cysts (right) assigned to particular cell types and time points from the embryo. Because the E8.75 gut tube has both embryonic and extraembryonic origin[49,50], it was not assigned to any of the two categories.

of cells from both Epi and VE cysts mapped to E7.5, indicating that cyst cells were developmentally more advanced than cells from BELAs. BELA cells mapped to both embryonic cell types (Epi, mesoderm (Mes), and germ cells) and extraembryonic cell types (PrE, embryonic VE (emVE), extraembryonic VE (exVE), parietal endoderm (ParE), and yolk sac endoderm (YsE)). Cells from Epi cysts in contrast mapped mostly to embryonic cell types, whereas cells from VE cysts mapped to extraembryonic cell types. The low number of cells from VE cysts that mapped to embryonic cell types likely originate from a small fraction of cells is refractory to PrE differentiation because of insufficient transgene induction levels[9]. The vast majority of embryonic cells from Epi cysts were labeled as mesoderm, whereas the embryonic cells from BELAs were labeled both as Epi and mesoderm. The extraembryonic cells from VE cysts mostly mapped to cell types that are not in contact with the epiblast, such as the exVE, the ParE, and the YsE, and only 4% mapped to the emVE. In BELAs in contrast, 8% of all cells were labeled as emVE. This corresponds to approximately one fifth of all extraembryonic cells in this sample, and indicates that the presence of the Epi core in BELAs promotes an emVE identity. We conclude that cells from all three 3D systems bear transcriptional similarity to the embryonic and extraembryonic lineages of the mouse embryo shortly after implantation. Furthermore, differences in developmental stage and cell type identity between embryonic cells from Epi cysts and BELAs, and between extraembryonic cells from VE cysts and BELAs, indicate that interactions between the two cell types regulate cell differentiation.

### Interaction of Epi and VE cells in BELAs promotes AVE differentiation

To characterize in more detail how interactions between Epi and VE cells in BELAs affect cell differentiation, we performed unsupervised clustering of the single-cell transcriptomes and searched for cell types that were present in BELAs, but not in the cyst samples. Epi cells clustered according to their sample of origin, with one cluster containing the Epi cells from BELAs (cluster 1), and the other one containing almost all Epi cyst cells (cluster 2, Fig. 3a, b). These global transcriptomic differences between Epi cells from the two sample types suggests that the signaling environment generated by the VE layer in BELAs differs from that generated by the artificial extracellular matrix used to grow Epi cysts. VE cells also fell into two clusters, but here, cells were not segregated based on their origin (Fig. 3a, b). Instead, cluster 3 consisted of both cells from VE cysts and BELAs, whereas a small cluster 4 contained exclusively cells from BELAs (Fig. 3a, b). Genes that were downregulated in cluster 4 mostly encoded components of the extracellular matrix (Supplementary Fig. 3 and Supplementary Data 1). The list of upregulated genes, on the other hand, contained transcription factors involved in AVE differentiation, such as Lhx1, Otx2, and Eomes (Fig. 3c and Supplementary Data 1). Furthermore, cells in cluster 4 specifically expressed the AVE markers Cer1, Lefty1, and Sfrp1, suggesting that they had adopted an AVE identity (Fig. 3c, d, Supplementary Data 1). To corroborate this finding, we integrated the transcriptomes of BELA cells from clusters 3 and 4 with an embryo data set that focused especially on the AVE differentiation from E5.5 to 6.25[15]. Consistent with the integration with the whole-embryo dataset above, the majority of cells from both cluster 3 and 4 mapped together with E6.25 embryo cells (79% and 76%, respectively, Fig. 3e–g). While 95% of cells from cluster 4 were labeled as AVE after integration, cells from cluster 3 were labeled both as AVE (39%) and Epi-VE (59%, Fig. 3g). This supports the notion of AVE differentiation in BELAs, and suggests that AVE gene expression signatures extend to cells beyond those identified in cluster 4. In contrast to integration with the whole-embryo dataset, where a large proportion of BELA-VE cells were mapped to the exVE lineage, virtually no cells obtained the corresponding ExE-VE label upon integration with the AVE-focused dataset. These discrepancies could be due to different

representations of the lineages, which may bias the outcome of the dataset integration, as well as diverging strategies for annotation in the two reference datasets: While Thowfeequ et al. separated ExE-VE from Epi-VE using an information theoretic criterion followed by annotation based on marker expression, Nowotschin et al. classified emVE and exVE cells according to their differentiation probabilities towards gut tube and yolk sac, respectively.

We next stained for the marker genes Gata6, Otx2, and Cer1 to validate the presence of AVE cells, and to determine how they are distributed amongst individual BELAs. In the embryo, Cer1 specifically marks the AVE, Otx2 is expressed in a broader domain encompassing the AVE and parts of the emVE[1], while Gata6 marks the entire VE, prompting us to use Otx2 and Cer1 expression as broad and specific AVE markers, respectively. In 28 out of 33 BELAs, we found cells that co-expressed OTX2 and GATA6 protein (Fig. 3h). Furthermore, such OTX2-positive cells were located outside the laminin-ring, as would be expected for an AVE identity (Fig. 3i). Similar results were obtained by in situ HCR staining for *Otx2* and *Cer1* mRNA (Fig. 3j, 27 out of 36 BELAs with cells co-expressing *Otx2* and *Gata6* mRNA; 10 out of 36 BELAs with cells co-expressing *Cer1* and *Gata6* mRNA). Light-sheet imaging of a Cer1:H2B-Venus transcriptional reporter[16] (Supplementary Fig. 4) integrated into our inducible lines indicated that AVE cells in BELAs tended to be spatially clustered (Fig. 3k, l, Supplementary Movie 4). AVE differentiation was specific to BELAs, as ExE embryoids made with X10 or IM8A1-GFP XEN cells did not have OTX2-expressing cells in their outer layer (Supplementary Fig. 5).

To test the functionality of the putative AVE, we adopted a protocol to trigger mesoderm differentiation in ESC aggregates[17], consisting of a 24 hour treatment with 3 µM Chir99021 (Chi) on day 2, followed by analysis 24 hours after the end of the Chi pulse (Supplementary Fig. 6a). Under these conditions, the fraction of BELAs containing cells expressing the mesoderm marker T/Bra was lower than that in EXE embryoids and Epi cysts (Supplementary Fig. 6b, c). While we cannot exclude the possibility that initial conditions impinge on mesoderm differentiation propensity – Epi cells in BELAs transited through an ICM-like state upon dox-treatment, while those in EXE embryoids and Epi cysts did not (Supplementary Fig. 6a) – this observation is consistent with the in vivo activity of the AVE to inhibit mesoderm differentiation. We did not detect a preferential relative orientation of the Cer1:H2B-Venus and T/Bra expression domains in BELAs after the Chi pulse (Supplementary Fig. 6d, e). This suggests that the positioning of the AVE and the site of mesoderm differentiation on opposite sides of the egg cylinder requires inputs that are not present in BELAs, such as localized mesoderm differentiation cues, or a boundary between epiblast and ExE cells.

In the embryo, AVE markers such as Otx2 and Cer1 cells are only transiently expressed between E5.5 and E7.5[14]. BELAs expressing these markers in the VE could first be detected 2 days after re-seeding, their number peaked at day 3, and declined thereafter (Fig. 3m), thus recapitulating the transient nature of the AVE in the embryo. Taken together, these results show that interactions between the Epi and the VE trigger AVE differentiation in small groups of spatially clustered cells in a large number of BELAs.

### Activin/Nodal signaling is necessary and sufficient for AVE differentiation

Next, we used LIANA[18], a ligand-receptor analysis framework, to identify potential Epi-derived signals that could trigger AVE differentiation in BELAs. Amongst the top scoring interactions between Epi and VE cells, we found ligand-receptor pairs associated with signaling from the extracellular matrix, FGF, and Eph-Ephrin signaling (Fig. 4a, Supplementary Data 2). Consistent with the critical role of Nodal signaling for AVE differentiation in the embryo[4], this analysis furthermore returned the Nodal receptors Acvr1b and Acvr2a and Nodal co-factor ligand Tdgf1. To test the function of Nodal signaling in BELAs, we used

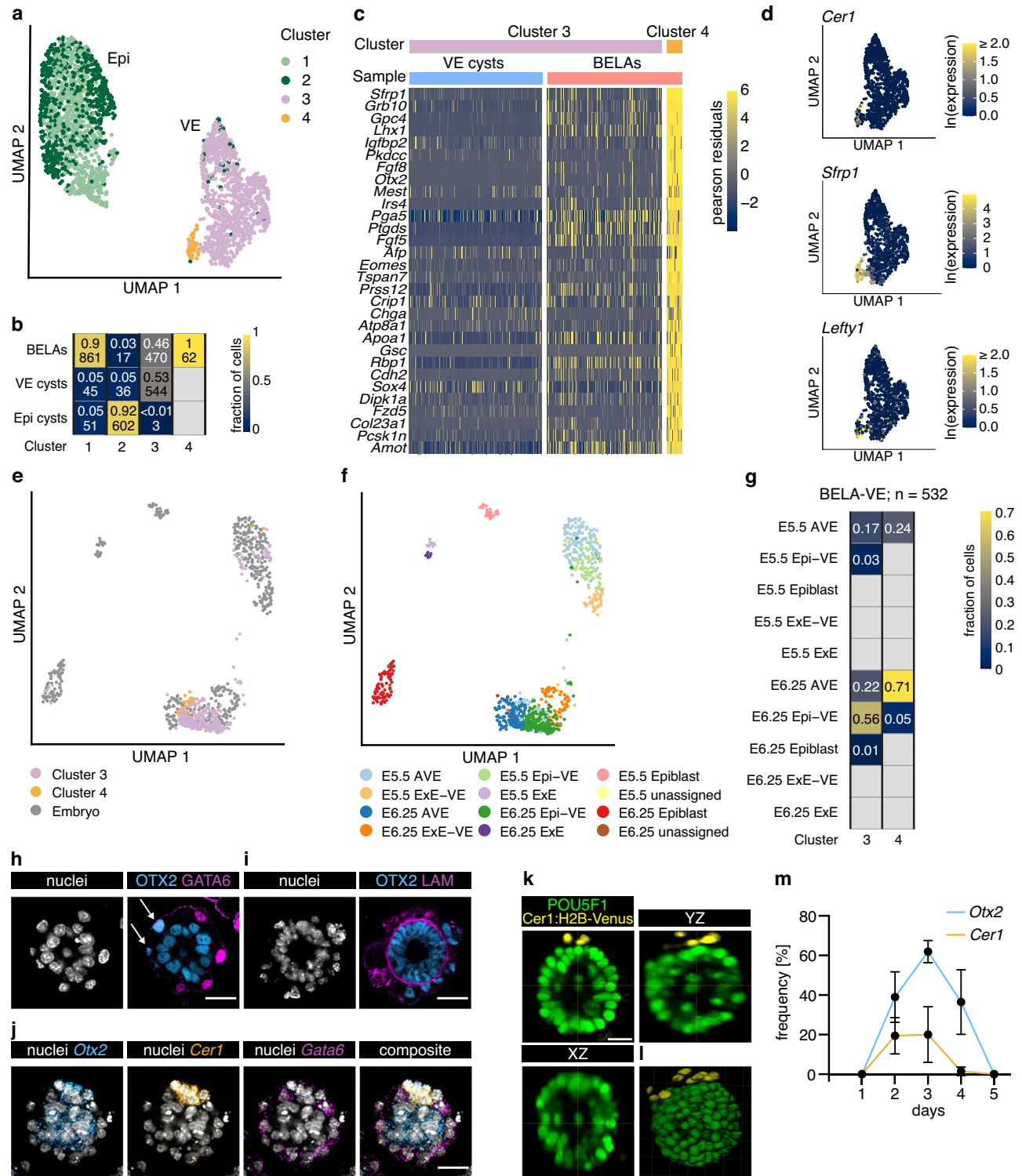

the receptor inhibitor SB431542 (SB43), and analyzed AVE differentiation in BELAs generated from *Nodal*-mutant cells. Both perturbations abrogated AVE differentiation, as judged by Otx2 and Cer1 expression (Fig. 4b–d). Epiblast-derived Nodal signals were required for AVE differentiation not only in BELAs but also in the embryo, since tetraploid complementation with Nodal mutant cells likewise resulted in the absence of a CER1-positive AVE (Supplementary Fig. 7).

We then asked whether Epi-derived Nodal signals were sufficient to trigger AVE differentiation in vitro. We again used an extended doxycycline pulse together with exogenous FGF4 to generate pure cultures of PrE cells, and seeded these on a high-adhesion substrate to

analyze differentiation in a homogeneous 2D VE-layer (Fig. 4e). Addition of the Nodal agonist ActivinA triggered the expression of both the Cer1:H2B-Venus reporter, as well as OTX2 expression, in a dose-dependent manner (Fig. 4f–i, Supplementary Fig. 8). Cultures were homogeneously GATA6-positive, both in the presence and absence of ActivinA (Supplementary Fig. 9). At 200 ng/ml ActivinA, approximately two thirds of all cells expressed OTX2, indicating that the majority of VE cells have AVE differentiation potential. ActivinA also triggered expression of EOMES, another critical factor for AVE differentiation[2] (Fig. 4j). At intermediate ActivinA concentrations we found that cells expressing AVE markers often occurred in spatial clusters, with a

**Fig. 3 | AVE differentiation in BELAs. a** UMAP representation of single-cell transcriptomes (same as in Fig. 2b), colored according to Louvain clustering. **b** Heatmap showing the fraction and total number of cells from each sample in the four clusters from **a**. The small number of cells from VE cysts in clusters 1 and 2 likely originate from cells that were refractory to PrE differentiation (see above, and Raina et al.[9]), **c** Heatmap showing the 30 most upregulated genes between the cells of cluster 3 and cluster 4 in (**a**), ordered by log2-fold change. Single-cell expression is shown as the Pearson residual of the normalized counts. **d** Zoom-in in UMAP from (**a**) showing expression of *Cer1*, *Sfrp1*, and *Lefty1* in VE cells. **e** UMAP of single-cell transcriptomes from BELA-VE cells (Cluster 3 and Cluster 4 in **a**, **b**), integrated with scRNAseq data from mouse embryos at E5.5 and E6.25 from[15]. **f** Same UMAP as in **e**, colored according to cell type annotation from[15] after integration and label transfer. **g** Heatmap showing the fraction of BELA-VE cells assigned to particular cell types and developmental time points from the embryo. **h** Immunostaining for the AVE marker OTX2 (blue) and the VE marker GATA6 (magenta). Arrows highlight co-expression. **i** Immunostaining for the AVE marker OTX2 (blue) and the basement membrane marker LAM (magenta). **h**, **i** one representative out of a total of at least 13 structures from *n* = 2 independent experiments shown. **j** In situ HCR staining for the AVE markers *Otx2* (blue) and *Cer1* (orange), and the VE marker *Gata6* (magenta). One representative out of a total of at least 60 structures from *n* = 3 independent experiments shown. **k**, **l** Orthogonal views (**k**) and 3D volume rendering (**l**) of a BELA stained for the Epi marker POU5F1 (green) and the AVE reporter Cer1:H2B-Venus (yellow) imaged with light sheet microscopy. One representative out of a total of 28 structures from *n* = 3 independent experiments shown. **m** Mean frequency of AVE marker gene expression in BELAs on different days after re-seeding. *Otx2* expression was scored as AVE marker only if it could clearly be assigned to the outer layer of BELAs. *n* = 2 independent experiments, for number of BELAs analyzed at each timepoint see Source Data file. Error bars indicate SD. Scale bars: 25 μm.

central Cer1:H2B-Venus/OTX2/EOMES triple-positive core, surrounded by cells that were OTX2 and EOMES single- or double-positive (Fig. 4f, j). This nested arrangement recapitulated the spatial arrangement of the Otx2 and Cer1 expression domains in the VE in the embryo and in BELAs (Fig. 4k–m). Taken together, these experiments show that Activin/Nodal signals are necessary and sufficient for the differentiation of AVE cells. The nested patterns of AVE marker expression in BELAs and the 2D VE layers furthermore point to the existence of tissue-intrinsic patterning mechanisms.

## β-catenin signaling restricts AVE differentiation to local cell clusters

We then wondered why AVE differentiation occurred in spatial clusters despite global stimulation with ActivinA in homogeneous 2D layers of VE cells. This could reflect clonal expansion of single cells that were privileged for AVE differentiation, or alternatively, be the consequence of local signaling domains that allow for AVE differentiation. To distinguish between these possibilities, we added three different fluorescent labels to the inducible cell lines and analyzed the clonal composition of Cer1:H2B-positive nests in mixed cultures (Fig. 5a, b). We found a similar number of nests that carried the same clonal label (13/30) and nests composed of cells with different labels (17/30, Fig. 5b). This suggests that the clonal expansion of single cells contributes to nest formation, but that in addition, local signaling environments generated by cell-cell communication promote AVE differentiation. In the embryo, BMP4 signals are thought to restrict differentiation[6]. We would, therefore, expect that activation of BMP signaling should block AVE differentiation in the 2D system, whereas BMP signaling inhibition should expand it. In contrast to this expectation, addition of BMP4 only mildly reduced the proportion of Cer1:H2B-Venus-positive cells and had no effect on OTX2 expression (Supplementary Fig. 10). Addition of the BMP receptor inhibitor LDN193189 did not increase the expression of either of the two markers (Supplementary Fig. 10). Therefore, BMP signaling does not play a strong role in restricting AVE differentiation in vitro. Since we noticed that Nodal was specifically expressed in AVE cells in the embryo[14,15] and in BELAs (Supplementary Fig. 11a), we tested whether Nodal autoregulation promoted AVE differentiation in nests. When we measured expression of AVE markers in *Nodal*-mutant cells, we found that the proportion of Cer1:H2B-Venus-positive cells was reduced by half compared to wild-type controls, but the proportion of OTX2-positive cells was unchanged (Supplementary Fig. 11b–f), and cells expressing the two markers were still spatially clustered (Supplementary Fig. 11d). Therefore, endogenous Nodal signaling plays a minor role in regulating AVE differentiation in vitro.

Finally, we asked which additional signaling system could be responsible for regulating AVE differentiation within the VE. The strong and specific expression of a TCF/LEF transcriptional reporter in the VE indicated β-catenin transcriptional activity in this tissue at peri-implantation stages[19] (Supplementary Fig. 12a). We therefore tested

the effects of the GSK3 inhibitor Chi, or the tankyrase inhibitor XAV993 (XAV), which will promote or inhibit β-catenin transcriptional activity, respectively. Chi treatment of embryos for 24 h from E5.25 onwards lead to an increase of TCF/LEF reporter activity (Supplementary Fig. 12a, b), suggesting that β-catenin activity can be further boosted in the VE at these developmental stages. Even though OTX2-levels appeared to be inversely correlated with TCF/LEF reporter expression (Supplementary Fig. 12b), we could not detect statistically significant differences in OTX2 expression between treated and control embryos. In this experiment, treatments were started immediately after egg cylinder formation, the earliest time point compatible with subsequent normal anterior-posterior axis patterning in control embryos. However, at this stage AVE differentiation has already begun, which may limit the effects of the inhibitors. We, therefore, turned to the 2D system, where inhibitors can be applied throughout the whole course of differentiation. Here, addition of Chi completely abrogated OTX2 and Cer1:H2B-Venus expression (Fig. 5c–f), but not GATA6 expression (Supplementary Fig. 9). XAV treatment, in contrast, triggered OTX2 expression in almost all cells and strongly boosted Cer1:H2B-Venus expression, while maintaining GATA6 expression (Fig, 5c, d, f, Supplementary Fig. 9). Inhibition of Wnt secretion with the porcupine inhibitor IWP2 also increased the expression of both Cer1:H2B-Venus and OTX2, but to a smaller extent than XAV (Fig. 5c–f), suggesting that β-catenin transcriptional activity in the VE might partially be regulated by Wnt-independent mechanisms. Taken together, the exogenous activation of Activin/Nodal signaling, combined with the inhibition of endogenous β-catenin transcriptional activity, allows the efficient differentiation of AVE cells following forced GATA expression in naïve pluripotent cells. The strong effects of β-catenin signaling manipulation on AVE differentiation furthermore suggest that the local inhibition of β-catenin transcriptional activity contributes to the formation of AVE nests.

## Discussion

Here, we report the differentiation of cohorts of AVE cells in bilayered embryo-like aggregates generated from mouse ESCs. We identify the underlying signaling events between embryonic and extraembryonic cells, and use this knowledge to develop a 2D AVE differentiation protocol. With this protocol, we demonstrate that an antagonism between tissue-intrinsic β-catenin transcriptional activity and Nodal signals coming from both the Epi and the AVE itself control AVE differentiation.

To investigate mechanisms of lineage crosstalk between the Epi and the VE, we used an experimental system where both lineages are established in reproducible proportions from a single starting population through cell-cell communication via FGF4[9]. This approach contrasts with previous studies, where bilayered aggregates have been formed by mixing ESCs with established XEN cell lines[11], by mixing wild-type ESCs with GATA-inducible ESCs[20], or by chemical conversion of ESCs towards the VE lineage[21]. Consistent

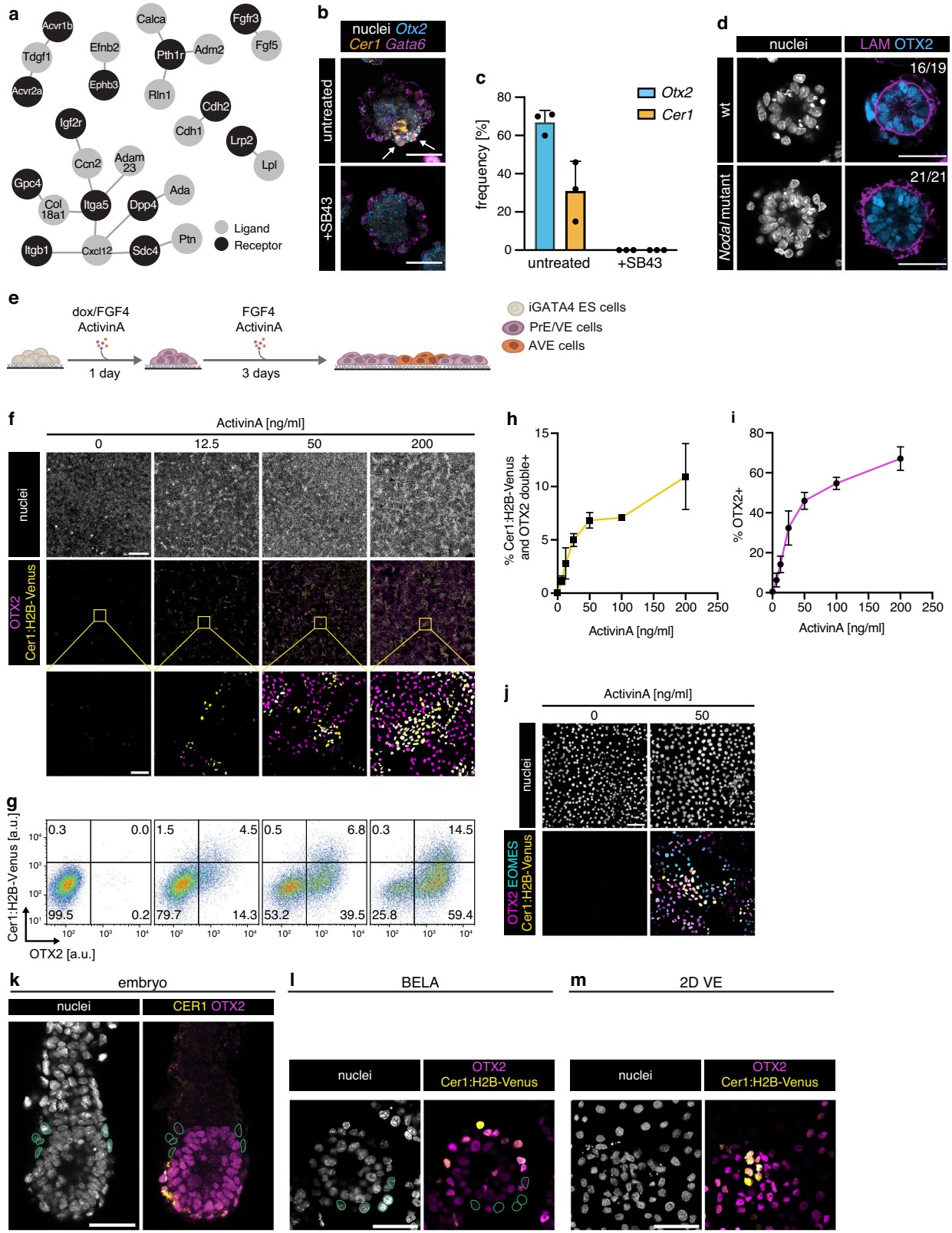

**f** ActivinA [ng/ml]

**h** % Cer1:H2B-Venus and OTX2 double+

**i** % OTX2+

**k** embryo

**l** BELA

**m** 2D VE

with our results, these previous studies found that the Epi core induces an embryonic identity in the overlying VE[21]. They also reported expression of the AVE marker Lefty1 in the VE[11,20], but whether these Lefty1-expressing cells had acquired an AVE identity remained unclear. Our scRNAseq analysis demonstrates that a subset of VE cells in BELAs differentiate into AVE, in contrast to XEN cells in EXE embryoids which seem to lack a comparable AVE differentiation potential. We speculate that AVE differentiation in BELAs benefits from the specification of the Epi and the VE from a single starting cell population, which closely recapitulates the situation in the embryo.

Using methods to fully direct ESC differentiation towards either Epi or VE, we were able to compare the behavior of pure populations of these lineages with that of mixed populations that form BELAs. In contrast to Epi cells that require exogenous extracellular matrix cues

**Fig. 4 | Activin/Nodal signaling is necessary and sufficient for AVE differentiation. a** Output of ligand-receptor analysis with LIANA[18], showing the top 20 interactions between Epi-derived ligands and VE-derived receptors. **b** In situ HCR staining of untreated (top) and SB43-treated (bottom) BELAs for AVE markers *Otx2* (blue) and *Cer1* (orange) and the PrE/VE marker *Gata6* (magenta). **c** Mean frequency of AVE marker gene expression in untreated and SB43-treated BELAs 3 days after re-seeding. Data in **b**, **c** from *n* = 3 independent experiments, one representative structure shown in **b**. Error bars in (**c**) indicate SD. **d** Immunostaining for LAM (magenta) and OTX2 (blue) in BELAs made from *Nodal* wild-type (top) and *Nodal*-mutant cells (bottom). At least 19 structures each from *n* = 3 independent *Nodal*-mutant clones were imaged, one representative structure shown. **e** Schematic of experimental protocol to generate 2D layers of VE cells for AVE differentiation. **f** Immunostaining for OTX2 (magenta) and H2B-Venus (yellow) of Cer1:H2B-Venus reporter cells treated with indicated concentrations of ActivinA for 3 days after an

extended doxycycline pulse. One out of *n* = 3 independent experiments shown. **g** Flow cytometry of cells differentiated and stained as in **f**. **h** Mean percentage of Cer1:H2B-Venus; OTX2 double-positive cells differentiated with increasing doses of ActivinA. *n* = 3 independent experiments, error bars indicate SD. **i** Same as **h** but showing percentage of OTX2-positive cells. **j** Immunostaining for OTX2 (magenta), EOMES (cyan), and H2B-Venus (yellow) of Cer1:H2B-Venus reporter cells treated as in **f**. One out of *n* = 2 independent experiments shown. **k–m** Immunostaining of a E5.5 mouse embryo (**k**, *n* = 3 embryos), a BELA (**l**, *n* = 21 BELAs), and cells in a 2D VE layer (**m**, *n* = 4 independent experiments) for OTX2 (magenta) and CER1 protein (**k**), or the Cer1:H2B-Venus reporter (**l**, **m**) (yellow). Green circles in **k**, **l** indicate OTX2-negative nuclei of cells in contact with the epiblast compartment. Scale bars: 50 µm in **b**, **d**, ((**f**) inset), (**j**), (**k–m**); 500 µm in **f**. Source data for **c**, **h**, **i** are provided in the Source Data file.

to form cysts[22], we find that pure cultures of VE cells spontaneously form cystic structures that resemble the outer layer of BELAs[23]. This finding suggests that the VE templates the formation of an organized Epi epithelium through the presentation of an extracellular matrix scaffold. Subsequent AVE differentiation in turn is dependent on the presence of the Epi core in BELAs. In line with previous studies from the embryo, we demonstrate that Epi-derived Activin/Nodal signals underlie this inductive event[4,24], thereby representing another swing of a pendulum of interactions between epiblast and VE[21].

Surprisingly, we find that AVE differentiation occurs in clusters of cells, both in BELAs and in the 2D differentiation protocol where ActivinA is applied globally. Current theories for AVE differentiation in the embryo posit that ExE-derived BMP signals restrict the differentiation of AVE precursors to VE cells at the distal tip of the egg cylinder[3,5,6]. However, the variable phenotypes upon loss of Bmp4 in embryos suggests that this signal is not solely responsible for the restriction of AVE differentiation, something that is further supported by the modest effects of BMP signaling activation and inhibition in our study. Our observation of restricted AVE differentiation in the absence of an ExE points to the existence of additional, tissue-intrinsic mechanisms that contribute to specifying AVE cells within the bulk of the VE. In the embryo, the cooperation of graded BMP signals with such tissue-intrinsic mechanisms may help to correctly position the AVE, and thereby confer robustness to axis patterning. In the future, it will be interesting to use simplified stem cell-based models such as BELAs to investigate how graded chemical and mechanical signals are integrated with tissue-intrinsic mechanisms for AVE differentiation.

The strong changes in AVE differentiation upon activating or blocking β-catenin transcriptional activity with small molecules identify β-catenin signaling as an important regulator of AVE differentiation. This idea is further supported by the expression of a TCF/LEF:H2B-GFP reporter throughout the VE[19,25], the specific expression of soluble Wnt inhibitors such as Sfrp1 and Sfrp5 in the AVE, as well as impaired AVE precursor differentiation in *Apc^{Min/Min}* embryos, where β-catenin is hyperactive[26]. We note that downregulation of Wnt/β-catenin signaling is also required for definitive endoderm differentiation[27], a lineage that bears transcriptional similarity to the AVE, thus pointing to a general role of Wnt/β-catenin dynamics in regulating endoderm differentiation.

In the VE, tissue-intrinsic Wnt/β-catenin signals may contribute to the patterned differentiation of AVE cells in local clusters. Similar Wnt-based patterning mechanisms underlie hair follicle differentiation in the mouse skin, and axis formation during planarian regeneration[28,29]. It is an attractive possibility that Wnt/β-catenin based patterning could underlie AVE differentiation and hence axis formation in disc-shaped non-rodent embryos that likely lack an external BMP gradient. Embryo models similar to BELAs but made from human ESCs[30] offer promising experimental tools to test this possibility, and to ultimately identify the components and the topology of the cell-cell communication network for AVE differentiation and patterning.

# Methods
## Cell lines
All embryonic stem cell lines used in this study were on an E14tg2a background[31]. The inducible Tet::GATA4-mCherry (iGATA) lines have previously been described[9]. We used two different clones in this study, C5 and C6, that differ in their induction rate. ESCs were maintained on fibronectin-coated dishes in an N2B27-based medium supplemented with 1 µM PD0325901 (SeleckChem), 10 ng/ml LIF (protein expression facility, MPI Dortmund), and 3 µM CHIR99021 (Tocris), referred to as 2i + LIF[32]. N2B27 was prepared as a 1:1 mixture of DMEM/F12 and Neuropan Basal Medium (both from PAN Biotech), supplemented with 1X N2 and 1X B27 supplements, 1X L-Glutamax, 0.0025% BSA, and 0.2 mM ß-mercaptoethanol (all from ThermoFisher). All iGATA4 cell lines were kept under constant selection with 200 µg/ml G418 (Sigma) to prevent silencing of the inducible transgene. The XEN cell lines IM8A1-GFP[33] and X10[34] have previously been described and were kindly shared by Kat Hadjantonakis. XEN cells were maintained in a GMEM-based medium, supplemented with 10% fetal bovine serum (FBS), 2 mM GlutaMAX, 1 mM sodium pyruvate, 0.1 mM β-mercaptoethanol and 10 ng/mL LIF. All cell lines were cultured at 37 °C with 5% $CO_2$, and regularly tested for mycoplasma contamination.

## Mouse strains
Animal experiments and husbandry were performed according to the German Animal Welfare guidelines and approved by the Landesamt für Natur, Umwelt und Verbraucherschutz Nordrhein-Westfalen (State Agency for Nature, Environment and Consumer Protection of North Rhine-Westphalia). The mice used in this study were at age from 6 weeks to 5 months. The animals were maintained under a 14-hour light/10-hour dark cycle with free access to food and water. Male mice were kept individually, whereas the female mice were housed in groups of up to four per cage.

Mice used for tetraploid complementation were of the B6C3F1 or CD1 strains and were raised in-house. Mice carrying the TCF/Lef:H2B-GFP reporter allele have previously been described[19]. To obtain TCF/Lef:H2B-GFP embryos, heterozygous TCF/Lef:H2B-GFP stud males were crossed with CD1 females.

## Generation of mutant and transgenic ESC lines
To generate a Cer1 reporter in the iGATA cell line, the Cer1 promoter region 4 kb upstream of the start codon was amplified from genomic DNA. A puromycin resistance cassette and a H2B-Venus sequence were amplified from Sprouty4 targeting vectors described in Morgani et al.[35], and Raina et al.[9]. All three fragments were cloned via Gibson assembly using a HiFi DNA assembly kit (NEB) into a vector backbone containing PiggyBac transposition sites[36]. The Cer1:H2B-Venus reporter construct was co-transfected with CAG-pBASE[36] using Lipofectamine 2000 (ThermoFisher) according to manufacturer's instructions. Cells were selected with 1.5 µg/ml puromycin (Sigma) starting 24 h after transfection. Colonies were picked one week after transfection,

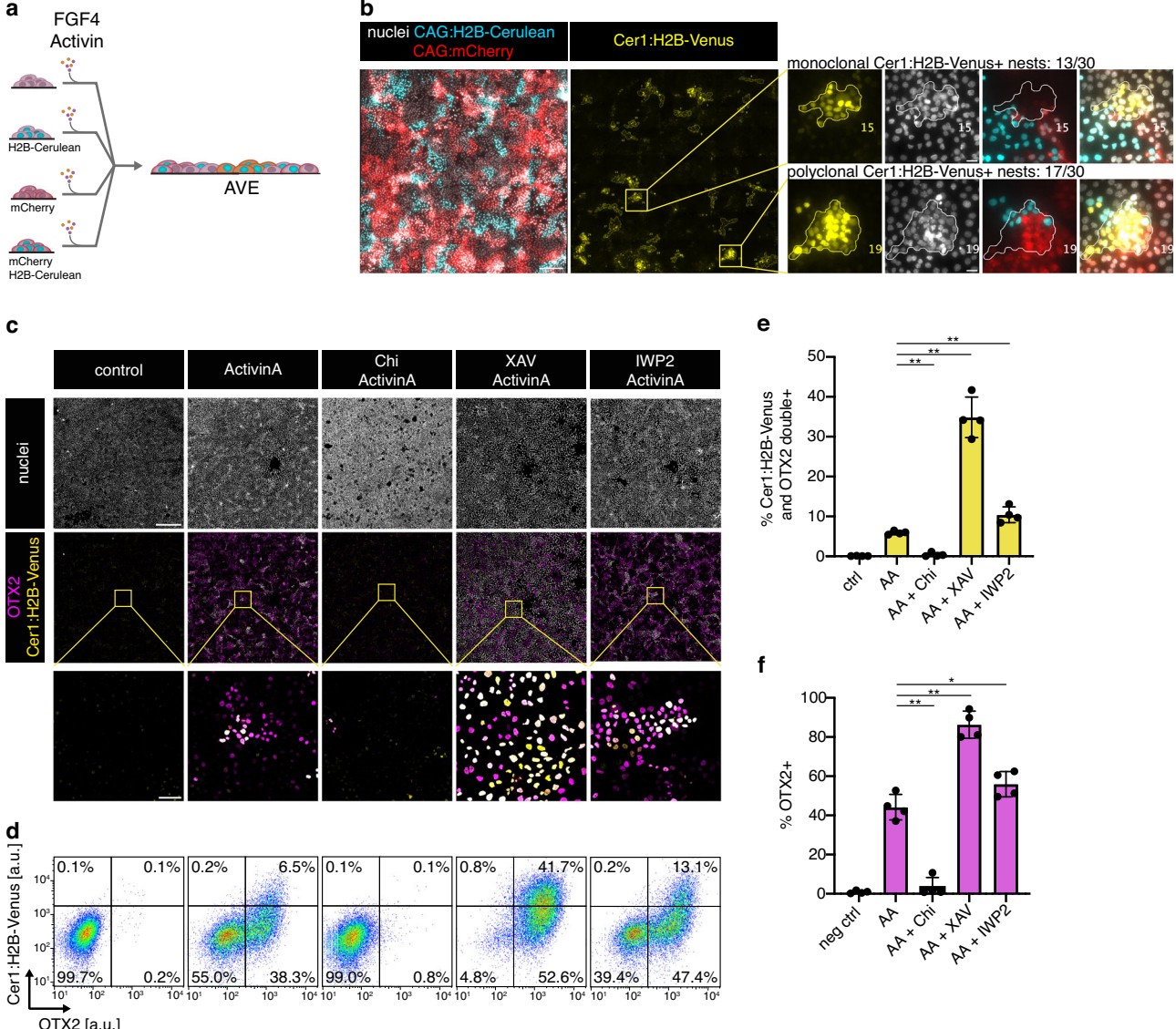

**Fig. 5 | Tissue-intrinsic β-catenin signals regulate AVE differentiation.**
**a** Experimental approach to determine clonal composition of AVE nests.
**b** Expression of clonal labels (red, cyan) and Cer1:H2B-Venus reporter (yellow) in cultures differentiated as in **a**. Insets on the right show examples of Cer1:H2B-Venus-expressing nests with a single clonal label (top, 13/30 nests), or with multiple labels (bottom, 17/30 nests). One out of $n = 2$ independent experiments shown.
**c** Immunostaining for OTX2 (magenta) and H2B-Venus (yellow) of Cer1:H2B-Venus reporter cells differentiated for 3 days after an extended doxycycline pulse with 50 ng/ml ActivinA (AA), together with 3 µM Chir99021 (Chi), 20 µM XAV939 (XAV), or 2 µM IWP2 as indicated. One out of $n = 3$ independent experiments shown. **d** Flow cytometry of cells differentiated and stained as in **c**. **e** Mean percentage of Cer1:H2B-Venus; OTX2 double-positive cells differentiated as in **c**. $n = 4$ independent experiments, error bars indicate SD. **p < 0.001 for AA vs. AA + Chi and AA vs. AA + XAV, and $p = 0.0045$ for AA vs. AA + IWP2 (two-tailed, unpaired $t$-test). **f** Same as **e** but showing percentage of OTX2-positive cells. **p < 0.001 for AA vs. AA + Chi and $p = 0.0001$ for AA vs. AA + XAV, *$p = 0.0434$ for AA vs. AA + IWP2 (two-tailed, unpaired $t$-test). Scale bars: 200 µm ((**b**) overview); 20 µm ((**b**) inset); 500 µm ((**c**) overview); 50 µm ((**c**) inset). Source data for **e**, **f** are provided in the Source Data file.

---

expanded, and evaluated for co-localization of Cer1 reporter activity and *Cer1* mRNA.

CRISPR/Cas9 was used to mutate the *Nodal* locus in iGATA ESCs (clone C6) and one subclone carrying the Cer1:H2B-Venus reporter construct. sgRNAs 5′-CCCCATGGACATACCCACTG-3′ and 5′-CCAGT CGAGCAGAAAAGTGT-3′ defining a 244 bp region in *Nodal* exon 2 were cloned into pX458 (Addgene plasmid #48138) or pX459 (Addgene plasmid #48139) using BbsI (NEB) according to Ran et al.[37]. Cells were transfected using Lipofectamine 2000 (Thermo Fisher Scientific) according to manufacturer's instructions. To enrich for transfectants, cells were either selected with 1.5 µg/ml puromycin for 2 days, or flow sorted for GFP-expression before seeding at clonal density. We established several clonal lines, and used primers 5′-GTGGACGTG ACCGGACAGAACTG-3′ and 5′-GGCATGGTTGGTAGGATGAAACTCC-3′

to PCR-amplify a sequence around the CRISPR mutation site. Clones that gave a shortened amplicon compared to the wild type were chosen for further analysis, and the exact sequence of the mutated alleles was determined by Sanger sequencing.

To generate constitutively labeled cell lines, we modified a pig-gybac vector for the constitutive expression of H2B-Cerulean[38] by either replacing its puromycin resistance cassette with a blasticidin resistance cassette from pCX-H2B-Cerulean-IRES-bsd[39] using restriction enzymes PmiI and PstI, or by replacing the H2B-Cerulean sequence with an mCherry coding sequence using restriction enzymes SpeI and NotI. Vectors were co-transfected with CAG-pBASE[36] using Lipofectamine 2000 according to the manufacturer's instructions, and transfected cells were selected with 15 µg/ml blasticidin 48 h after transfection. Four days after transfection, cells were flow sorted

for the expression of fluorescent proteins, and seeded a clonal density. Several clones were expanded, and two to three suitable clones with homogeneous, moderate H2B-Cerulean and/or mCherry fluorescence were selected by epifluorescence microscopy for further experiments.

### Differentiation of pure cultures of PrE cells and subsequent AVE differentiation

Pure cultures of PrE cells from iGATA clone C6 were obtained by inducing with 0.5 μg/ml doxycycline in 2i + LIF for 8 h, followed by another 16 h of doxycycline treatment in N2B27 supplemented with 10 ng/ml FGF4 and 1 μg/ml heparin. To obtain these cultures from iGATA clone C5, a 4 h pulse with 0.5 μg/ml doxycycline in 2i + LIF followed by further culture in N2B27 supplemented with FGF4 and heparin was sufficient. Clone C5 was used for the data shown in Fig. 1k–m and Supplementary Fig. 2c, in all other instances, clone C6 was used.

To differentiate AVE cells from these cultures, cells were additionally treated with 50 ng/ml ActivinA upon media change from 2i + LIF to N2B27. Approximately 24 h after the start of doxycycline treatment, cells were re-seeded at a total density of 25,000 to 30,000 cells/ cm² on fibronectin-coated dishes and cultured for 3 days in N2B27 supplemented with 10 ng/ml FGF4, 1 μg/ml heparin and 50 ng/ ml ActivinA.

### Formation of EXE embryoids, BELAs, VE- and Epi-cysts

EXE embryoids were generated by seeding ESCs and XEN cells at a ratio of 30:70 and a final density of 30,000 cells/cm² on dishes that had been coated with 0.1% gelatin in PBS for 30 min, followed by culture in N2B27 for 3 or 4 days.

BELAs were generated by inducing iGATA ESCs with 0.5 μg/ml doxycycline in 2i + LIF for 8 h, followed by a media change to N2B27 for 16 h. Cells were then seeded at a density of 30,000 cells/cm² on dishes that had been coated with 0.1% gelatin in PBS for 30 min. Floating aggregates were collected for further analysis at indicated time points.

VE cysts were generated from pure cultures of PrE cells differentiated as described above, followed by re-seeding onto gelatin-coated dishes at a density of 30,000 cells/cm² in N2B27 medium supplemented with 10 ng/ml FGF4 and 1 μg/ml heparin.

Cysts of Epi cells were made according to Bedzhov and Zernicka-Goetz, 2014[22], with minor modifications. iGATA ESCs were detached, resuspended in growth factor-reduced matrigel (Corning) and plated as 25 μl drops on μ-slides (ibidi). The slides were incubated at 37 °C to allow the matrigel to solidify and then filled with prewarmed N2B27 or 2i + LIF medium.

### Generation of epiblast-specific Nodal knock-out embryos

Epiblast-specific *Nodal* knock-out embryos were generated via tetraploid complementation. Donor embryos used for tetraploid complementation were derived from the B6C3F1 strain and foster mothers for embryo transfer experiments were from the CD1 background. Briefly, tetraploid morulae were aggregated with *Nodal*-mutant ESCs[4,38] or wild-type E14 ESCs. The aggregated embryos were cultured in KSOM (Millipore) for an additional 3 days, which were then transferred into the uterus of foster mothers. Post-implantation embryonic day (E) 5.5 tetraploid embryos were recovered by manually dissecting the uterus.

### Embryo culture

The E5.5 embryos were cultured in a 4-well plate (176740, Thermo Fisher) for 24 h, in IVC (50% IVC1 and 50% IVC2) medium, at 37 °C, 5% CO2 atmosphere in air. Before use, the IVC1 and IVC2 medium was equilibrated for 30 min at 37 °C, 5% CO₂ atmosphere in air. In experiments involving inhibitors treatments, the IVC medium was supplemented with pharmacological compounds: 3 μM CHIR99021 (Cat# 4423, Tocris) or 20 μM XAV (Cell Guidance Systems, Cat# SM38-10).

The composition of the IVC1 medium was previously described[22,40], consisting of DMEM F-12 (21331-046, Invitrogen), supplemented with 20% heat-inactivated FCS (10828028, Invitrogen), 0.5x Pen (25 U/ml) / Strep (25 μg/ml) (P4333, Sigma), 2 nM ʟ-glutamine (G7513, Sigma), 1x ITS-X (51500-056, Invitrogen), 8 nM β-estradiol (E8875, Sigma), 200 ng/ml Progesterone (P0130, Sigma) and 25 μM *N*-acetyl-ʟ-cysteine (A7250, Sigma). The IVC2 medium[22,40] was slightly modified, consisting of DMEM (12800017, Thermo Fisher Scientific) supplemented with 1.0 g/l NaHCO3 (S5761, Sigma Aldrich) 5% heat-inactivated FCS (10828028, Invitrogen), 30% (vol/vol) KSR (10828028, Invitrogen), 0.5x Pen (25 U/ml)/Strep (25 μg/ml) (P4333, Sigma), 2 nM ʟ-glutamine (G7513, Sigma), 1x ITS-X (51500-056, Invitrogen), 8 nM β-estradiol (E8875, Sigma), 200 ng/ml Progesterone (P0130, Sigma), 25 μM *N*-acetyl-ʟ-cysteine (A7250, Sigma).

### Immunostaining

BELAs and VE cysts in suspension were fixed with 4% paraformaldehyde at room temperature for 1 h, washed 5 times with phosphate-buffered saline (PBS) for 5 min each, and then incubated in PBS supplemented with 1% BSA and 0.1% Triton X-100 (PBT + BSA) for 3 h at room temperature, followed by incubation with primary antibodies diluted in PBT + BSA at 4 °C overnight. Primary antibodies used were anti-Oct3/4 (POU5F1, Santa Cruz Biotechnology sc-5279 1:100), anti-E-Cadherin (CDH1, Takara M108, 1:200), anti-GATA6 (R&D, RF1700, 1:200), anti-CD29 (ITGB1, BD Pharmingen 562153, 1:100), anti-LAM (Sigma L9393, 1:500 − 750), anti-OTX2 (Neuromics GT15095, 1:200), anti-pERM (Cell Signaling Technology #3141, 1:200), anti-PODXL (R&D MAB1556, 1:200), anti-SOX17 (R&D AF1924, 1:200), anti-ZO-1 (Invitrogen 61-7300, 1:100), and anti-GFP (Abcam ab13970 1:200).

To remove the primary antibody solution, the aggregates were washed five times with PBT + BSA. Aggregates were then incubated overnight at 4 °C with secondary antibodies diluted in PBT + BSA. Secondary antibodies from Invitrogen/Life Technologies were Alexa Fluor-conjugated and used at 4 μg/ml. Nuclei were stained with Hoechst 33342 dye at 1 μg/ml (Invitrogen). The secondary antibody solution was removed by 5 washes with PBS supplemented with 0.1% Triton X-100. The aggregates were resuspended in PBS and mounted onto μ-slides (ibidi).

Epi cysts in matrigel and cells grown in μ-slides (ibidi) were stained similarly, but with extended incubation and wash times for Epi cysts, and shortened times for cells grown as 2D layers. Samples were mounted in mounting solution consisting of 16% PBS, 80% glycerol, and 4% n-propyl-gallate.

Post-implantation embryos were either fixed immediately after isolation, or after culture as described above, in 4% PFA for 20 min, and washed twice in wash buffer containing 1% fetal calf serum (FCS) in PBS. The embryos were then permeabilized in 0.1 M glycine/0.3% Triton-X in PBS for 10 min, and washed twice in the wash buffer. The embryos were then incubated with primary antibodies in blocking buffer containing 2% FCS in PBS overnight at 4 °C. Primary antibodies used were anti-Cer1 (rat monoclonal, R&D systems, Cat# MAB1986, 1:200), anti-Otx2 (goat polyclonal, R&D systems, Cat# AF1979, 1:200), anti-Oct-4A (rabbit monoclonal (D6C8T), Cell signalling, Cat# 83932 S, 1:200), and anti-GFP (chicken polyclonal, Abcam ab13970, 1:200). After two washes in wash buffer, embryos were incubated with secondary antibodies and DAPI (Carl Roth, Cat# 6335.1) in blocking buffer, which were washed twice on the next day. The stained embryos were mounted in droplets of wash buffer on 35 mm μ-dish glass bottom plates (ibidi), covered with mineral oil and stored at 4 °C until imaging.

### In situ HCR

For third generation in situ HCR we used probe sets, wash and hybridization buffers together with corresponding Alexa Fluor-labeled amplifiers from Molecular Instruments[41]. Staining was performed according to the manufacturer's instructions. Briefly, samples were

fixed for 15 min to 1 h with 4% paraformaldehyde, washed four times with PBS with 0.1% Tween 20 (PBST) and permeabilized at least overnight in 70% ethanol at −20 °C. Samples were then washed twice with PBST, and equilibrated in probe hybridization buffer for 30 min at 37 °C. Transcript-specific probes for *Otx2* (NM_144841.5)*, Gata6* (NM_010258) and *Cer1* (NM_009887.2) were designed by Molecular Instruments. Probes were used at a final concentration of 4 nM in probe hybridization buffer and incubated overnight at 37 °C. To remove the probe solution, the sample was washed four times with probe wash buffer preheated to 37 °C and once with 5x SSC with 0.1% Tween 20 (SSCT). Samples were then equilibrated in amplification buffer for 30 min at room temperature. Alexa Fluor-labeled amplifiers were used at a final concentration of 60 nM together with Hoechst 33342 dye at 1 μg/ml and incubated overnight at room temperature. The amplifier solution was removed by six washes with 5x SSCT. Stained BELAs were resuspended in PBS and mounted on an ibidi μ-slide for imaging. 2D cultures were mounted in mounting solution consisting of 16% PBS, 80% glycerol, and 4% n-propyl-gallate.

## Imaging

Cells for long-term imaging (Fig. 1b and Supplementary Fig. 1b) were seeded at a density of 30,000 cells/cm$^2$ on 6-well plates (Sarstedt) or 8-well μ-slides plates (ibidi) and allowed to attach for 1–2 h before the start of imaging. Time-lapse movies were recorded with a 20×0.5 NA air objective on an Olympus IX81 widefield microscope equipped with a stage top incubator (ibidi), LED illumination (pE4000, CoolLED) and a c9100-13 EMCCD (Hamamatsu) camera. Hardware was controlled by MicroManager software[42], and tile scans were stitched in FIJI using the pairwise stitching plugin[43]. Live VE cysts in Fig. 1i, j were imaged on a Leica DM IRB widefield microscope using a 20×0.4 NA (Fig. 1i) or a 40×0.55 NA (Fig. 1j) phase contrast objective.

Stained BELAs, embryos, and stained cells in 2D culture were imaged on a Leica SP8 confocal microscope (Leica Microsystems) with a 63×1.4 NA oil immersion objective.

Cultures to determine the clonal composition of AVE clusters in 2D culture (Fig. 5b) were fixed, incubated with SYTO Deep Red Nucleic Acid Stain (ThermoFisher) for one hour, and imaged with a 20×0.5 NA air objective on an Olympus IX81 widefield microscope equipped with LED illumination (pE4000, CoolLED) and an iXon 888 EM-CCD camera (Andor). Hardware was controlled by MicroManager software[42] and tile scans were stitched in FIJI using the pairwise stitching plugin[43].

For light sheet imaging, fixed and stained aggregates were resuspended in low melting agarose and placed in 1.5 mm U-shaped capillaries (Leica). Capillaries were placed into water filled 35 mm high glass bottom μ-dishes (ibidi). Images were acquired using an HC Fluotar L 25×0.95 NA water DLS TwinFlect 2.5 mm detection objective and an HC PL Fluotar 5×0.15 NA illumination objective on a Leica TCS SP8 digital light sheet microscope. 3D animations were created using the Leica X application suite. Z-Stack images were processed and quantified using FIJI and Imaris.

## Analysis of spatial patterning in BELAs

To determine if mesoderm differentiation in BELAs was spatially patterned by Cer1:H2B-Venus expressing AVE cells, we calculated polarization vectors of the Cer1:H2B-Venus and T/Bra expression domains in BELAs according to Simunovic et al.[44], We first selected BELAs for imaging that contained T/Bra+ cells one day after a 24-hour Chi pulse. Mean polarization vectors were calculated in 2D for two independent Z-slices in each BELA that were 15 μm apart in Z-direction to ensure that an independent set of cells was analyzed. In each slice, we selected Epi and VE cells based on a mask that was drawn along the Laminin-ring that separates the two compartments. Next, we segmented individual nuclei with the StarDist 2D plugin in FIJI, using the versatile (fluorescent nuclei) model with default post-processing parameters[45]. Under- and oversegmented cells, debris and segmentation artefacts at

the boundaries of the Epi- and the VE-compartment were filtered out with size and circularity filters. We measured median fluorescence intensities per nucleus in each channel, and rescaled intensities in individual nuclei by the maximum per-nucleus median intensity in the same channel. Using this information on nuclei position and fluorescence intensity, we calculated the radius of gyration as a measure for structure size separately for each compartment, as well as the mean polarization vectors of Cer1:H2B-Venus and T/Bra expression normalized by $R_{gyr}^2$ according to Simunovic et al.[44]. To estimate background polarization vectors in the absence of patterning, we performed randomizations per Z-slice and BELA, where we shuffled measured fluorescence intensities between nuclei positions and determined the average polarization vector of 100 randomizations for each BELA and Z-slice. This analysis revealed that not all BELAs showed a polarized Cer1:H2B-Venus domain, possibly due to the Chiron treatment which is expected to inhibit AVE differentiation, and the late stage of analysis. We categorized BELAs as AVE-polarized, when the normalized polarization vector of the Cer1:H2B-Venus domain was larger than the polarization vector in 95% of the shuffled control group. Finally, we calculated the angle between the mean polarization vector of the Cer1:H2B-Venus and the T/Bra-domains for AVE-polarized and non-polarized BELAs.

## Flow cytometry

Cells for flow cytometry were detached from culture vessels, fixed in 4% paraformaldehyde for 15 min, washed with PBS and then incubated in PBS + 1% BSA + 0.25% Saponin (PBSap) for 30 min at room temperature. Afterwards, cells were incubated with primary antibodies diluted in PBSap at 4 °C overnight. The next day, cells were washed three times in PBSap and incubated with secondary antibodies diluted in PBSap for at least one hour. Cells were washed three times in PBSap, and passed through a cell strainer and analyzed immediately on a LSRII flow cytometer (BD Biosciences). Live cells were sorted on a FACS Aria Fusion (BD Biosciences). Flow cytometry data was analyzed with FlowJo (BD Biosciences).

## ScRNAseq sample preparation

BELAs and VE cysts were generated as described above. Between 100 and 200 BELAs and VE cysts were manually picked under a dissection microscope for further processing. We selected round aggregates and cysts, and excluded structures that contained a large number of dead cells, or that were unusually big or small. For the VE cysts, we also aimed at excluding structures that contained a clearly visible core of putative Epi-like cells, which likely arise from insufficiently induced cells. Both BELAs and VE cysts were gently spun down, resuspended in 1 ml Accutase and incubated at 37 °C for 10 min, followed by mechanical dissociation by pipetting and further incubation in Accutase for 5 min. Next, cells were spun down, washed in PBS, and resuspended in a small volume of PBS + 0.5% BSA.

To generate Epi cysts for RNA sequencing, single cells were seeded in matrigel and cultured in 2i + LIF for one day. Then, medium was changed to N2B27, and cells were cultured for another 3 days. Cysts were recovered from matrigel by incubation in recovery solution (Corning) for 20 min on ice. Next, cysts were gently spun down and dissociated with Accutase as described for BELAs and VE cysts above. To remove residual matrigel, dissociated cells were washed once with recovery solution and twice with ice-cold PBS, followed by resuspending in a small volume of PBS + 0.5% BSA.

## ScRNAseq library preparation and sequencing

Cells from all three samples were counted, and each sample was mixed with H$_2$O and RT master mix from the Chromium Next GEM Single Cell 3' GEM, Library & Gel Bead Kit v3.1 (10x Genomics) to obtain a cell density required for targeting 1000 (Epi and VE cysts) or 2000 (BELAs) cells. Cell suspensions were loaded on a Chromium Controller (10x Genomics) to

partition cells with gel beads in emulsion. Reverse transcription, cDNA recovery and amplification, and sequencing library construction were performed according to manufacturer's instructions (10x Genomics ChromiumNextGEMSingleCell_v3.1_Rev_D). We chose 12 PCR cycles for cDNA amplification, and 13 PCR cycles for index PCR. Concentration and insert size of sequencing libraries were determined with a BioAnalyzer High Sensitivity DNA Assay (Agilent). Libraries were sequenced by paired-end Illumina sequencing on a NovaSeq6000 instrument with a read length of 150 bp. We first performed sequencing at shallow depth with a target of 10.000 reads per cell, to confirm capturing of an appropriate number of high-quality single-cell transcriptomes. Subsequently, deeper sequencing was performed, to obtain between 100,000 and 150,000 reads per cell.

### ScRNAseq data analysis

Demultiplexing, alignment to the mouse genome mm10 (GENCODE vM23/Ensembl 98, from 10x Genomics) and read quantification was performed with CellRanger (10x Genomics, v4.0.0). Subsequent analysis was carried out in R using Seurat v4.1.1[46]. We first filtered out cells with less than 4000 different features detected and with more than 10% of the reads mapped to mitochondrial genes. SCTransform[46] was used to normalize and scale the molecular count data. For Uniform Manifold Approximation and Projection (UMAP) representation and clustering, shared cell populations were matched across samples using Seurat's integration algorithm for SCTransformed data with reciprocal PCA to identify anchors. Differentially expressed genes between the clusters resulting from Louvain clustering were identified with the FindMarker function based on the SCTransform normalized data, and sorted by fold-change.

ScRNAseq data from the developing mouse embryo was obtained from two publications: Raw counts of the E3.5 to E8.75 embryo dataset from Nowotschin et al.[14], including cell type annotations were downloaded from https://endoderm-explorer.com. For visualization, we did not differentiate between the different types of gut tube cells annotated by Nowotschin et al.[14], but used "gut tube" as a single label for all these cells. Similarly, we did not differentiate between different samples collected from E8.75 embryos, but pooled these groups with a single E8.75 label. This dataset was integrated with all single-cell transcriptomes from our study in SCANPY[47], using log1p-transformed counts after normalization of our data to 10,000 reads per cell. The asymmetric integration and label transfer was performed with ingest and cell type proportions were visualized in R using a custom heatmap function based on pheatmap.

ScRNAseq data and annotations of an embryo dataset focused on AVE development was obtained from the authors[15]. Integration of this dataset was performed with BELA cells from clusters 3 and 4 in Fig. 3a only, using the same pipeline as for the Nowotschin dataset.

### Cell-cell communication analysis

For the inference of cell-cell communication events from scRNAseq data we used LIANA, a LIgand-receptor ANalysis frAmework[18]. To identify cell-cell communication events in BELAs, we only used transcriptomes from this sample, and grouped them into two lineages according to the clustering in Fig. 3a: All cells from clusters 1 and 2 were grouped as Epi, and cells from clusters 3 and 4 were grouped as VE. The consensus database for ligand-receptor interactions was matched to its mouse ortholog genes using the omnipath database, and interactions were ordered by their consensus rank obtained from LIANA. For Fig. 4a, the top 20 interactions were displayed as an undirected adjacency graph.

### Statistics and reproducibility

Quantitative data are represented as mean ± SD. n in figure legends refers to number of biological replicates if not indicated otherwise. Where representative examples such as micrographs are shown, the number of independent experiments and number of structures imaged is indicated in the figure legends. For numbers of independent samples analyzed for each timepoint and condition in Figs. 3m and 4c please refer to the Source Data file. For flow cytometry experiments in Figs. 4 and 5 and Supplementary Figs. 10 and 11, at least $n = 20,000$ cells were analyzed for each condition in each biological replicate. Statistical analysis was performed in R or in GraphPad Prism 8 (v8.4.3), using unpaired or paired ratio t-tests as indicated in the figure legends. The significance of differential gene expression between clusters in scRNAseq data was assessed with a Wilcox likelihood-ratio test in R.

### Reporting summary

Further information on research design is available in the Nature Portfolio Reporting Summary linked to this article.

## Data availability

Single-cell RNA-sequencing data generated in this study has been deposited at the NCBI gene expression omnibus repository under accession number GSE198780. Source data are provided with this paper.

## Code availability

All code used for analysis and visualization, together with a list of the R packages used, is available on GitHub at https://github.com/Schroeterlab/BELAs_Schumacher_et_al. A permanent version of the code has been published with doi: 10.5281/zenodo.11102648 on Zenodo[48]. Any additional information required to reanalyze the data reported in this paper is available from the authors upon request.

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

## Acknowledgements

We thank M. Sandhaus for his contributions to the early stages of this project, Sarah Teckhaus for help with generating Nodal mutant cells, and Pauliine Konsa for pilot experiments on mesoderm differentiation in BELAs. We are grateful to Shankar Srinivas and Antonia Scialdone for sharing unpublished sequencing data. We thank P. Bastiaens, current and former members of the Schröter group, and all members of the Department for Systemic Cell Biology for support, stimulating discussions, and conceptual input on the project, as well as Alfonso Martinez Arias, Nicholas Rivron, Stefan Semrau, Naomi Moris, and Vikas Trivedi for feedback on earlier versions of this manuscript. This work was supported by the VW foundation (project no. A130140 "OntoTime"), the German Research foundation (project no. 441798639), the German Center for the protection of laboratory animals (Bf3R project no. 1328-567), an ERC consolidator grant (MORPHEUS, 101043753 to I.B.) and the Max Planck Society.

## Author contributions

Conceptualization and methodology: S.S. and C.S.; Validation: S.S. and M.M.; Formal analysis: S.S. and M.F.; Investigation: S.S., M.F., M.M., R.C.,

Y.S.K., and C.S.; Data curation: M.F.; Writing – original draft: S.S., M.F., and C.S.; Writing – review and editing: all authors; Visualization: S.S. and M.F.; Supervision: I.B. and C.S.; Funding acquisition: I.B. and C.S.

## Funding

## Competing interests

The authors declare no competing interests.
