## [Peer Review File · Nature Communications]

REVIEWER COMMENTS

Reviewer #1 (Remarks to the Author):

Anterior-posterior patterning and the gradual induction of the AVE in the normal mouse embryo is known to depend on the integration of numerous signaling pathways emanating from all three primary lineages of the post-implantation embryo. Confusingly most of the key ligands and signaling pathway components are expressed in multiple tissues, underscoring the importance of reciprocal positive and negative feedback loops that control dose-dependent patterning of the early embryo. For example Nodal and Wnt3 are both expressed in the epiblast and overlying VE. Bmp4 is expressed in the ExE, while Bmp2 is expressed in the VE and posterior epiblast.

Loss of function and conditional genetic experiments have highlighted the complex temporal and spatial signaling requirements for A-P patterning and AVE specification. Loss of function of Bmp4 results in a spectrum of embryonic defects ranging from a block at very early gastrulation stages to embryos that gastrulate normally but die at early somite stages, suggesting that Bmp4 signaling per se is not essential for A-P patterning and AVE induction (Winnier et al, 1995). Bmp2 is expressed in both the epiblast and VE and Bmp2 deletion results in complex phenotypes (Gavrilov and Lacy, 2013). Nodal null embryos show no evidence of A-P patterning (Brennan et al, 2001). Nodal signaling from the the epiblast is essential to activate the Smad2 dependent pathway in the VE to allow AVE induction (Lu and Robertson 2004). Wnt3 null embryos arrest at post-implantation stages and show no evidence for overt A-P patterning and AVE induction (Liu et al, 1999). Selective removal of Wnt3 activity from the epiblast or VE results in very different phenotypes highlighting the interplay of Wnt3 signaling between these two tissue populations (Barrow et al, 2003; Tortelote et al, 2013; Yoon et al, 2015). However it is evident that Wnt activity in the epiblast is dispensable for induction of AVE markers and A-P patterning (Tortelote et al, 2013) pointing to a critical role for Wnt signaling within the VE. Finally as yet unidentified signals originating from the ExE are required to restrict DVE/AVE induction to the distal-most VE since physical removal of the ExE results in a remarkable expansion of AVE marker gene expression (Rodriguez et al, 2005).

In the present paper Schumacher and colleagues employ a previously described in vitro system exploiting ESC harboring a dox-inducible Gata4 expression cassette to generate different tissue “structures” to further investigate tissue interactions that pattern the Epi versus VE. The experiments compare three different types of “structures” namely simple EBs (which they term BELAs) comprised of an inner layer of Epiblast overlain with VE, cysts of Epi tissue alone, or large VE alone cysts generated via Gata4 over-expression in the presence of exogenous Fgf4 (Figure 1). It is not clear what advantages these so-termed BELAs offer as compared to the EXE-embryoids described by Zhang et al (Nat Comms 2019)? (See Zhang Fig 1&2). EXE-embryoids also generate random patches of AVE-like cells as assessed by expression of Lefty1-cherry and single cell PCR analysis of Cer1 and Otx2 expression (Zhang Fig 5).

They use sc-seq approaches (Figure 2) to characterize the cell populations that form in these three structures, utilizing sc-seq embryo data sets published by the Hadjantonakis lab. They show that the EB/BELAs are the most complex in terms of the sub-sets of VE generated, while the Epi balls express mostly mesoderm markers and the VE alone are comprised mostly of ExVE. They conclude that, as is known from the embryo, signaling interactions between the Epi and VE in the EB/BELA structures underlie robust generation of EmbVE sub-sets.

Next (Figure 3) they further informatically sub-divide the VE populations that form in the EBs into discrete clusters. They identify a cluster (Cluster 4) with proposed AVE identity based on the upregulated expression of *Lhx1*, *Otx2*, *Eomes* and *Gsc*. However they fail to appreciate that many of these markers identify EmbVE – while *Eomes*, *Otx2* and *Lhx1* have roles in DVE/AVE formation, they are broadly expressed throughout the EmVE prior to the onset of gastrulation. For example *Otx2*, used in the present study as an AVE marker, is dynamically expressed in the EmbVE and not exclusive to the DVE or AVE (as reported in Acampora et al, 1998, Acampora et al, 1999, Hoshino et al, 2015). Next they attempt to further corroborate that Cluster 4 cells represent AVE by integrating the EB/BELA data set with a second sc-seq embryo data set published on BioRxiv (reference is cited as both 16 and 18 in the manuscript). However as the authors themselves state on lines 229-232 this reveals some major discrepancies which they attribute to potentially divergent strategies for annotation between the 2 data sets. It is therefore unclear if the Cluster 4 cells have a normal embryonic counterpart. Using IHC and various staining approaches to examine co-expression of GATA6 with either *Otx2* (note this is not an AVE marker per se) or *Cer1* they conclude that putative AVE cells are found as small patches within the outer VE layer of the EB/BELAs. A similar finding was reported by Zhang et al using EXE-embryoids and a *Lefty1*-cherry reporter.

They next re-examine the requirement for Nodal signaling in AVE induction (Figure 4). Using the well characterized receptor inhibitor SB431542 as well as Nodal mutant ESCs they show that blocking Nodal expression prevents the expression of *Otx2* and *Cer1* in EB/BELAs. Furthermore they exploit Nodal mutant ESCs to generate 4N chimeric embryos in which the epiblast is Nodal deficient – accordingly these embryos show no evidence for AVE induction. These findings corroborate previously published reports (Brennan et al, 2001; Lu and Robertson 2004) and confirm that Nodal signaling from the epiblast is essential for activating the Smad2 dependent pathway known to be essential for DVE/AVE induction in the normal embryo. Not surprisingly addition of exogenous activin to VE populations generated via *Gata4* over-expression results in the upregulation of the Smad2 target gene *Eomes* (Simon et al, 2017) as well as expression of *Otx2* and *Cer1*, which are often co-expressed in small clusters of cells within the VE monolayer.

Finally (Figure 5) they address the possible underlying mechanism that leads to the formation of these clusters of putative AVE cells within VE cell layers. Lineage tracing approaches show they arise by local clonal expansion. Unsurprisingly addition of exogenous BMP or the BMP inhibitor

LDN193189 fails to affect Cer1 or Otx2 expression. They also exploit in vitro generated Nodal deficient VE to show as previously (Liu and Robertson, 2004) that endogenous Nodal activity within the VE does not play a major role in AVE induction. Finally they manipulate the Wnt pathway via addition of the agonist Chir99021 - which results in a failure to upregulate Otx2 and Cer1, or the Wnt inhibitors XAV939 or IWP2 which result in upregulation of Otx2 and Cer1. Collectively these in vitro data show that within the VE lineage exogenous provision of Nodal signaling together with abrogation of Wnt signaling allows the formation of nests of AVE-like cells.

Collectively the in vitro experiments contained in the manuscript reinforce the widely accepted scenario that in the embryo Nodal signaling from the epiblast promotes a sub-set of the VE to activate Wnt and Nodal inhibitors (Cer1, Dkk, Sfrp5, mfz8, Lefty1 etc) which are necessary to allow the formation of the AVE within the EmbVE. Moreover as shown in numerous studies, the forming DVE/AVE population is very finely patterned (e.g. Hoshino et al, 2015). The ESC derived models used in the current study exclude the ExE known to provide as yet unidentified "signals" that constrain DVE/AVE formation to the distal tip of the embryo. While the current studies have been carefully carried out, and a large body of experimental data is included, unfortunately no new information has emerged with regard as to how the dose-dependent integration of signals between the primary embryonic lineages leads to highly localized patterning of the VE that ultimately results in induction of the DVE. It remains unclear if ESC-derived in vitro models can accurately mimic the complex temporal and spatial signaling environment that emerges in the context of the post-implantation embryo. The recent development of synthetic embryo structures derived from mixing cell lines from all three primary cell lineages (e.g. Zhang et al, 2019) may afford a more robust system to address the fine tuning of the pathways governing AVE induction and localization.

Reviewer #2 (Remarks to the Author):

The authors use novel stem cells based embryo models to gain insight into the mechanisms necessary for AVE induction and geometrical restriction. The data confirm the interplay between epiblast and VE for AVE differentiation as well as the prominent role for Nodal in that process, and identify Wnt as an intrinsic factor that antagonises AVE differentiation.

The models are very powerful as they allow differentiation and organisation of the two germ layers from a single ESC population, thereby reducing a series of confounding factors and allowing reliable identification of molecular and cellular pathways involved in epiblast and VE morphology (for example the VE cyst with the inverse apicobasal polarity is surprising and interesting) as well as their interaction, and finally AVE differentiation and arrangement in clusters. The authors make a great use of available single cell RNA Seq data (notably from Thowfeequ et al.), and provide experiments in mouse embryo to back up some of the in vitro findings.

The experiments are solid, the presentation precise and beautiful, and the manuscript very well written.

-On a conceptual point of view, the statement concerning the non-requirement of the BMP gradient in vivo may be a bit too strong. The fact that an mouse AVE-like population can arise without ExE ectoderm may be another example of the robustness of the developmental processes, not a demonstration that it is indeed not necessary for optimal embryo geometry. In other contexts it was shown that the gradient itself was determinant, so it could be interesting to reproduce a graded signaling in vitro, or perhaps this could be discussed in the manuscript.

-To strengthen the relevance of the presented data in other species, would it be possible to test the inhibitors used on the stem cells models on rabbit embryos ?(although I admit that this could be out of the scope of this particular manuscript). This may disturb other processes, but it is conceivable that application of a drug for a short amount of time in the right developmental window could provide additional insight.

-An interesting finding is the transcriptomic similarity of the stem cells based model for epiblast to mesoderm. This opens the possibility to explore the functional role of the AVE like cells generated in the model. For example, is there an inverse geographical correlation between the Otx2/Cer positive VE like cells and the expression of mesoderm markers in the internal layer of BELAs?

Response to reviewers comments Schumacher et al. (Nature Communications manuscript NCOMMS-23-37886)

We would like to thank the reviewers for their comments and constructive criticism on our manuscript. Below we provide a point-by-point response to explain the additional data and changes to the manuscript that we have made to address the reviewers' comments. Text of the original reviews is in black, our responses are in blue. In the revised manuscript, all changes to the text are tracked in red font.

Reviewer #1 (Remarks to the Author):

Anterior-posterior patterning and the gradual induction of the AVE in the normal mouse embryo is known to depend on the integration of numerous signaling pathways emanating from all three primary lineages of the post-implantation embryo. Confusingly most of the key ligands and signaling pathway components are expressed in multiple tissues, underscoring the importance of reciprocal positive and negative feedback loops that control dose-dependent patterning of the early embryo. For example Nodal and Wnt3 are both expressed in the epiblast and overlying VE. Bmp4 is expressed in the ExE, while Bmp2 is expressed in the VE and posterior epiblast.

Loss of function and conditional genetic experiments have highlighted the complex temporal and spatial signaling requirements for A-P patterning and AVE specification. Loss of function of Bmp4 results in a spectrum of embryonic defects ranging from a block at very early gastrulation stages to embryos that gastrulate normally but die at early somite stages, suggesting that Bmp4 signaling per se is not essential for A-P patterning and AVE induction (Winnier et al, 1995). Bmp2 is expressed in both the epiblast and VE and Bmp2 deletion results in complex phenotypes (Gavrilov and Lacy, 2013). Nodal null embryos show no evidence of A-P patterning (Brennan et al, 2001). Nodal signaling from the the epiblast is essential to activate the Smad2 dependent pathway in the VE to allow AVE induction (Lu and Robertson 2004). Wnt3 null embryos arrest at post-implantation stages and show no evidence for overt A-P patterning and AVE induction (Liu et al, 1999). Selective removal of Wnt3 activity from the epiblast or VE results in very different phenotypes highlighting the interplay of Wnt3 signaling between these two tissue populations (Barrow et al, 2003; Tortelote et al, 2013; Yoon et al, 2015). However it is evident that Wnt activity in the epiblast is dispensable for induction of AVE markers and A-P patterning (Tortelote et al, 2013) pointing to a critical role for Wnt signaling within the VE. Finally as yet unidentified signals originating from the ExE are required to restrict DVE/AVE induction to the distal-most VE since physical removal of the ExE results in a remarkable expansion of AVE marker gene expression (Rodriguez et al, 2005).

We thank the reviewer for nicely putting our work into context. We believe that the reviewer has made an error in their reference to the work by Liu et al., 1999 (PMID: 10431240). This paper shows that the AVE forms normally in Wnt3 null embryos (see figures 4u – x in PMID: 10431240), arguing against the reviewer's assertion that critical roles for Wnt signaling within the VE have previously been known.

Furthermore, we would like to point out that BMP4 has been suggested to constitute the signal originating from the ExE that restricts DVE/AVE induction (Yamamoto et al., 2009 (PMID: 19153222), Soares et al., 2005 (PMID: 16381610)). However, as the reviewer rightly mentions, the broad spectrum of phenotypes upon loss of Bmp4 suggests that this signal is not solely responsible for restriction of AVE differentiation. We now explicitly discuss this point in lines 477 - 480 of the revised manuscript. In the light of these previous results from the embryo, our work therefore makes an important contribution towards improving models for AVE differentiation and axis patterning.

In the present paper Schumacher and colleagues employ a previously described in vitro system exploiting ESC harboring a dox-inducible Gata4 expression cassette to generate different tissue “structures” to further investigate tissue interactions that pattern the Epi versus VE. The experiments compare three different types of “structures” namely simple EBs (which they term BELAs) comprised of an inner layer of Epiblast overlain with VE, cysts of Epi tissue alone, or large VE alone cysts generated via Gata4 over-expression in the presence of exogenous Fgf4 (Figure 1). It is not clear what advantages these so-termed BELAs offer as compared to the EXE-embryoids described by Zhang et al (Nat Comms 2019)? (See Zhang Fig 1&2). ExE-embryoids also generate random patches of AVE-like cells as assessed by expression of Lefty1-cherry and single cell PCR analysis of Cer1 and Otx2 expression (Zhang Fig 5).

To address the reviewer’s question what advantages BELAs offer as compared to the EXE embryoids described by Zhang et al. (PMID: 30700702), we have generated aggregates that resemble EXE embryoids by mixing ESC and two commonly used XEN cell lines under the same conditions as used for preparation of BELAs. These new results are shown in the new Supplementary Figs. 1, 2e, and 5, and are described in lines 96 – 105, 127 – 129, and 263 – 265 of the revised manuscript. Briefly, we find that our EXE embryoids and BELAs have an overall similar size and architecture, but in contrast to BELAs, EXE embryoids do not detach from the surface of the culture vessel. Furthermore, in our hands, the outer XEN cell-derived layer in EXE embryoids lacked strong CDH1 expression as well as a continuous apical domain marked by pERM and PODXL that was commonly seen in BELAs. This suggests that the ESC-derived VE cells in BELAs have a stronger propensity to epithelialize than two commonly used XEN cell lines, a difference that is further supported by the inability of XEN cells to spontaneously form VE cysts. Finally, we show that OTX2 protein as a marker for emVE differentiation can be readily detected in the outer layer of BELAs, but not in EXE embryoids.

Taken together, these new results indicate that the ESC-derived VE cells in BELAs have a higher differentiation potential than the established XEN cells that we used to form EXE embryoids, which represents an important advantage when it comes to investigating mechanisms of extraembryonic lineage differentiation. We acknowledge that differentiation potential may differ between XEN cell lines, thus offering an explanation why Zhang et al., who used an independently derived XEN cell line, detected evidence for emVE differentiation in EXE embryoids using a Lefty1-cherry reporter and single cell PCR analysis of Cer1 and Otx2 expression.

They use sc-seq approaches (Figure 2) to characterize the cell populations that form in these three structures, utilizing sc-seq embryo data sets published by the Hadjantonakis lab. They show that the EB/BELAs are the most complex in terms of the sub-sets of VE generated, while the Epi balls express mostly mesoderm markers and the VE alone are comprised mostly of ExVE. They conclude that, as is known from the embryo, signaling interactions between the Epi and VE in the EB/BELA structures underlie robust generation of EmbVE sub-sets.

Next (Figure 3) they further informatically sub-divide the VE populations that form in the EBs into discrete clusters. They identify a cluster (Cluster 4) with proposed AVE identity based on the upregulated expression of *Lhx1*, *Otx2*, *Eomes* and *Gsc*. However they fail to appreciate that many of these markers identify EmbVE – while *Eomes*, *Otx2* and *Lhx1* have roles in DVE/AVE formation, they are broadly expressed throughout the EmVE prior to the onset of gastrulation. For example *Otx2*, used in the present study as an AVE marker, is dynamically expressed in the EmbVE and not exclusive to the DVE or AVE (as reported in Acampora et al, 1998, Acampora et al, 1999, Hoshino et al, 2015).

We thank the reviewer for pointing out that the expression domains of *Eomes*, *Otx2* and *Lhx1* in the embryo extend beyond the DVE/AVE. We now explicitly acknowledge the broader expression of *Otx2* compared to *Cer1* (lines 252 - 255, and new Figure 4k). Prompted by the reviewer's comment, we looked at expression of additional, more specific AVE markers, such as *Lefty1* and *Sfrp1* (new panel d in Fig. 4). These markers are also specifically expressed in cluster 4, in line with our original conclusion that cells in this cluster have adopted an AVE identity.

Next they attempt to further corroborate that Cluster 4 cells represent AVE by integrating the EB/BELA data set with a second sc-seq embryo data set published on BioRxiv (reference is cited as both 16 and 18 in the manuscript). However as the authors themselves state on lines 229-232 this reveals some major discrepancies which they attribute to potentially divergent strategies for annotation between the 2 data sets. It is therefore unclear if the Cluster 4 cells have a normal embryonic counterpart.

To address the reviewer's concern, we now describe the diverging strategies for annotation in the reference datasets in detail (lines 243 - 249): While Thowfeequ et al. separated ExE-VE from Epi-VE using an information theoretic criterion followed by annotation based on marker expression, Nowotschin et al. classified emVE and exVE cells based on their differentiation probabilities towards gut tube and yolk sac, respectively. Importantly, these differences concern the separation between emVE and exVE identities, and not the question whether cells acquire an AVE identity (the reference dataset by Nowotschin et al. does not contain an AVE annotation). We therefore do not believe that the discrepancies between our the two integrations question that cells in cluster 4 are most similar to the AVE of the peri-implantation embryo.

The accidental double referencing of the bioRxiv study by Thowfeequ et al. has been corrected.

Using IHC and various staining approaches to examine co-expression of GATA6 with either Otx2 (note this is not an AVE marker per se) or Cer1 they conclude that putative AVE cells are found as small patches within the outer VE layer of the EB/BELAs. A similar finding was reported by Zhang et al using EXE-embryoids and a Lefty1-cherry reporter.

Prompted by the reviewer's comment to Fig. 1 (advantage of BELAs compared to ExE embryoids) we have compared OTX2 protein expression as a broad AVE marker between BELAs and ExE embryoids, and found that it can be readily detected in the outer layer of BELAs, but not in EXE embryoids formed with two commonly used XEN cell lines (new Supplementary Fig. 5). Although this discrepancy between the study by Zhang et al. and our work may be explained by a higher differentiation potential in the XEN cell lines used by Zhang et al. compared to the ones used by us, these data emphasize the importance of using appropriate, differentiation-competent cell lines for investigating extraembryonic lineage differentiation.

They next re-examine the requirement for Nodal signaling in AVE induction (Figure 4). Using the well characterized receptor inhibitor SB431542 as well as Nodal mutant ESCs they show that blocking Nodal expression prevents the expression of Otx2 and Cer1 in EB/BELAs. Furthermore they exploit Nodal mutant ESCs to generate 4N chimeric embryos in which the epiblast is Nodal deficient – accordingly these embryos show no evidence for AVE induction. These findings corroborate previously published reports (Brennan et al, 2001; Lu and Robertson 2004) and confirm that Nodal signaling from the epiblast is essential for activating the Smad2 dependent pathway known to be essential for DVE/AVE induction in the normal embryo. Not surprisingly addition of exogenous activin to VE populations generated via Gata4 over-expression results in the upregulation of the Smad2 target gene Eomes (Simon et al, 2017) as well as expression of Otx2 and Cer1, which are often co-expressed in small clusters of cells within the VE monolayer.

We agree with the reviewer that the finding that ActivinA treatment induces Eomes expression is on its own not surprising. We wish to emphasize though that the results of this figure go beyond this observation in two important aspects. First, we report and validate a strategy to differentiate VE and AVE from ESCs in vitro. To our knowledge, such a protocol has not been published before. Since it will open new avenues for investigating the signaling control of this important differentiation step in mammalian development, this represents a significant advance. Still more important is our observation of the spatially patterned expression arrangement of different AVE markers in the 2D setting, which resembles their arrangement in the mouse embryo as well as in BELAs. This observation points to a tissue-intrinsic mechanism of AVE differentiation and patterning that we further explore in Fig. 5. To emphasize this important message from the figure, we have now added stainings of E5.5 embryos and BELAs that show the similarities in the marker expression domains between the three systems (new panels k – m), and discuss these similarities in the text in lines 341 - 343. To make space for these new data, we have relegated the panels showing the absence of an AVE in 4N chimeric embryos with a Nodal-deficient

epiblast to a new Supplementary Fig. 7, because, as pointed out by the reviewer, these results corroborate previously published reports.

Finally (Figure 5) they address the possible underlying mechanism that leads to the formation of these clusters of putative AVE cells within VE cell layers. Lineage tracing approaches show they arise by local clonal expansion. Unsurprisingly addition of exogenous BMP or the BMP inhibitor LDN193189 fails to affect Cer1 or Otx2 expression. They also exploit in vitro generated Nodal deficient VE to show as previously (Liu and Robertson, 2004) that endogenous Nodal activity within the VE does not play a major role in AVE induction. Finally they manipulate the Wnt pathway via addition of the agonist Chir99021 - which results in a failure to upregulate Otx2 and Cer1, or the Wnt inhibitors XAV939 or IWP2 which result in upregulation of Otx2 and Cer1. Collectively these in vitro data show that within the VE lineage exogenous provision of Nodal signaling together with abrogation of Wnt signaling allows the formation of nests of AVE-like cells.

We would like to thank the reviewer for their comment. Regarding our finding that “addition of exogenous BMP or the BMP inhibitor LDN193189 fails to affect Cer1 or Otx2 expression”, we would like to point out that BMP4 has previously been identified as an inhibitory signal from the ExE that restricts DVE formation (Yamamoto et al., 2009, PMID: 19153222). We would therefore expect that activation of BMP signaling should block AVE differentiation in the 2D system, whereas BMP signaling inhibition should expand it, but this is not what we find. We now explicitly state this expectation in lines 379 - 381, to emphasize that the weak effects of BMP signaling manipulation are indeed a surprising result.

Furthermore, we decided to replace “Wnt signaling” with “ β -catenin activity” in the title and in the relevant sections throughout the manuscript, to capture our observation that manipulations to β -catenin transcriptional activity through the small molecules Chir99021 or XAV, give consistently stronger phenotypes than manipulations affecting Wnt ligand activity through the small molecules IWP-2. This suggests that β -catenin activity, and not necessarily Wnt, antagonizes Nodal-driven AVE differentiation.

Collectively the in vitro experiments contained in the manuscript reinforce the widely accepted scenario that in the embryo Nodal signaling from the epiblast promotes a sub-set of the VE to activate Wnt and Nodal inhibitors (Cer1, Dkk, Sfrp5, mfz8, Lefty1 etc) which are necessary to allow the formation of the AVE within the EmbVE. Moreover as shown in numerous studies, the forming DVE/AVE population is very finely patterned (e.g. Hoshino et al, 2015). The ESC derived models used in the current study exclude the ExE known to provide as yet unidentified “signals” that constrain DVE/AVE formation to the distal tip of the embryo. While the current studies have been carefully carried out, and a large body of experimental data is included, unfortunately no new information has emerged with regard as to how the dose-dependent integration of signals between the primary embryonic lineages leads to highly localized patterning of the VE that ultimately results in induction of the DVE. It remains unclear if ESC-derived in vitro models can accurately mimic the complex temporal and spatial signaling environment that emerges in the context of the post-

implantation embryo. The recent development of synthetic embryo structures derived from mixing cell lines from all three primary cell lineages (e.g. Zhang et al, 2019) may afford a more robust system to address the fine tuning of the pathways governing AVE induction and localization.

We thank the reviewer for acknowledging the careful execution of our experiments. With regard to their suggestion to use more complex in vitro models, we would like to stress that the main characteristic of our approach is to exploit the modularity of stem cell-based systems to reduce the complexity of cell-cell interactions. It is this approach that first of all demonstrates the existence of a tissue-intrinsic mechanism that can direct AVE differentiation in the absence of ExE-derived signals, as is evident in the fine patterning of the AVE in our simplified 2D system. We emphasize this important finding in lines 345 and 346 of the revised manuscript. Second, our approach leads to the identification of β -catenin signaling as a novel player in AVE differentiation. We therefore believe that both our approach contributes to expanding the knowledge in the field.

Reviewer #2 (Remarks to the Author):

The authors use novel stem cells based embryo models to gain insight into the mechanisms necessary for AVE induction and geometrical restriction. The data confirm the interplay between epiblast and VE for AVE differentiation as well as the prominent role for Nodal in that process, and identify Wnt as an intrinsic factor that antagonises AVE differentiation.

The models are very powerful as they allow differentiation and organisation of the two germ layers from a single ESC population, thereby reducing a series of confounding factors and allowing reliable identification of molecular and cellular pathways involved in epiblast and VE morphology (for example the VE cyst with the inverse apicobasal polarity is surprising and interesting) as well as their interaction, and finally AVE differentiation and arrangement in clusters. The authors make a great use of available single cell RNA Seq data (notably from Thowfeequ et al.), and provide experiments in mouse embryo to back up some of the in vitro findings. The experiments are solid, the presentation precise and beautiful, and the manuscript very well written.

We thank the reviewer for their positive comments on the overall design of our study, and in particular for highlighting the power of our experimental system. We believe that the reviewer's remark that "differentiation and organisation of the two germ layers from a single ESC population ...[reduces]...a series of confounding factors" further strengthens our point that BELAs represent several advantages over previously described systems, such as the EXE embryoids mentioned by reviewer #1.

-On a conceptual point of view, the statement concerning the non-requirement of the BMP gradient in vivo may be a bit too strong. The fact that an mouse AVE-like population can arise without ExE ectoderm may be another example of the robustness of the developmental processes, not a demonstration that it is indeed not necessary for optimal embryo geometry. In other contexts it was shown that the

gradient itself was determinant, so it could be interesting to reproduce a graded signaling in vitro, or perhaps this could be discussed in the manuscript.

We agree with the reviewer that our data do not argue against the role of the BMP gradient in the embryo. To avoid creating this impression, we have re-written the relevant sections in the abstract (lines 25 - 29) and the discussion (lines 481 - 485). Specifically, we have removed any statements that suggest the BMP gradient is not necessary for AVE differentiation and patterning. Instead, we now spell out the hypothesis that the coupling of our newly discovered tissue-intrinsic mechanism with external gradients may confer robustness to axis patterning.

Whether graded signaling could direct the site of AVE differentiation in vitro is an interesting question. However, we believe that such experiments are beyond the scope of the current manuscript. As suggested by the reviewer, we therefore discuss this possibility as a goal for future studies in lines 485 - 487 of the discussion section.

-To strengthen the relevance of the presented data in other species, would it be possible to test the inhibitors used on the stem cells models on rabbit embryos?(although I admit that this could be out of the scope of this particular manuscript). This may disturb other processes, but it is conceivable that application of a drug for a short amount of time in the right developmental window could provide additional insight.

We agree that it would be highly interesting to explore the signaling control of AVE differentiation in embryos of other species, such as the rabbit. However, as already acknowledged by the reviewer, these types of experiments are extremely demanding and require access to specialized resources, which is why they are out of the scope of the present manuscript.

Still, to address the reviewer's point we have made use of a TCF/LEF reporter mouse strain that reports transcriptional activity of β -catenin in the VE at peri-implantation stages. In these embryos, we observe an increase of reporter expression upon treatment with Chi99021, as well as an inverse correlation between OTX2 expression and reporter intensity. These results are in line with a potential role of β -catenin signaling in AVE differentiation inferred from our in vitro system. These new data are now presented in a new Supplementary Fig. 12, and described in the text in lines 396 - 411.

-An interesting finding is the transcriptomic similarity of the stem cells based model for epiblast to mesoderm. This opens the possibility to explore the functional role of the AVE like cells generated in the model. For example, is there an inverse geographical correlation between the Otx2/Cer positive VE like cells and the expression of mesoderm markers in the internal layer of BELAs?

Prompted by the reviewer's suggestion, we have included in the revision a set of experiments to test the functionality of the AVE-like cells in the model. Because the number of cells expressing mesoderm markers such as T/Bra in BELAs was low on day 4 of BELA formation when an AVE region was still present, we decided to boost

mesoderm differentiation by a 24-hour pulse with the Wnt agonist Chi99021, adopting a protocol that is widely used to trigger mesoderm differentiation in ESC aggregates termed gastruloids. We first compare the extent of mesoderm differentiation in BELAs with that in EXE embryoids (which are surrounded by a VE layer but do not have an overt AVE; see new Supplementary Fig. 5) and Epi cysts, which lack a VE layer altogether. Consistent with a possible functional role of the AVE cells in BELAs, we find that the fraction of BELAs containing T/Bra+ cells is lower than that in EXE embryoids and Epi cysts. Focusing on those BELAs that do contain T/Bra+ cells, we determined the polarization vectors of the Cer1:H2B-Venus+ and T/Bra+ domains and the angle between them. This analysis revealed that the orientation of a T/Bra+ domain was not strongly constrained by the position of a Cer1:H2B-Venus domain. These results suggest that the reciprocal positioning of an AVE and the site of mesoderm differentiation requires inputs that are not present in BELAs, such as localized mesoderm differentiation cues, or a boundary between epiblast and ExE cells. These new results are shown in a new Supplementary Fig. 6, and described in the text in lines 266 - 281.

REVIEWERS' COMMENTS

Reviewer #1 (Remarks to the Author):

The authors have provided a robust response to the original reviewers comments and have revised the manuscript and provided additional data accordingly. I have no further comments that require attention prior to acceptance.

Reviewer #2 (Remarks to the Author):

The authors have answered my concerns adequately.